# Accelerating MHC-II Epitope Discovery via Multi-Scale Prediction in Antigen Presentation

## Abstract

Antigenic epitope presented by major histocompatibility complex II (MHC-II) proteins plays an essential role in immunotherapy. However, compared to the more widely studied MHC-I in computational immunotherapy, the study of MHC-II antigenic epitope poses significantly more challenges due to its complex binding specificity and ambiguous motif patterns. Consequently, existing datasets for MHC-II interactions are smaller and less standardized than those available for MHC-I. To address these challenges, we present a well-curated dataset derived from the Immune Epitope Database (IEDB) and other public sources. It not only extends and standardizes existing peptide–MHC-II datasets, but also introduces a novel antigen–MHC-II dataset with richer biological context. Leveraging this dataset, we formulate three major machine learning (ML) tasks of peptide binding, peptide presentation, and antigen presentation, which progressively capture the broader biological processes within the MHC-II antigen presentation pathway. We further employ a multi-scale evaluation framework to benchmark existing models, along with a comprehensive analysis over various modeling designs to this problem with a modular framework. Overall, this work serves as a valuable resource for advancing computational immunotherapy, providing a foundation for future research in ML guided epitope discovery and predictive modeling of immune responses.

## 1 Introduction

The major histocompatibility complex (MHC), including both Class I (MHC-I) and Class II (MHC-II) proteins, is essential for immune surveillance. Among them, MHC-II-mediated antigen presentation is particularly crucial. Antigenic epitopes are bound to MHC-II and presented on the surface of antigen-presenting cells (APCs), where they are then recognized by $CD4^+$ T-cells to initiate immune responses or maintain self-tolerance (Ishina et al., 2023). Recently, emerging researches further highlight the importance of MHC-II epitopes in cancer immunotherapy, where they can directly stimulate $CD4^+$ T-cells and indirectly affect $CD8^+$ T-cell responses (Alspach et al., 2019; Brightman et al., 2023).

Despite these promising roles, MHC-II epitope discovery remains considerably unexplored, especially within computational frameworks. A substantial gap exists between models developed in this domain and the broader advances in machine learning (ML). We believe the reasons are three-fold: (1) MHC-II interactions are inherently challenging to model, as the highly polymorphic alleles exhibit an open binding groove that accepts peptides of variable lengths, making the binding patterns more complicated. (2) Available experimental datasets for MHC-II interactions are smaller, noisier, more unbalanced, and less standardized than the MHC-I counterparts (Reynisson et al., 2020; Vita et al., 2018), which introduces additional challenges for robust ML development. (3) The problem is less exposed to the ML community, such that the most acknowledged and widely used methods to date remain simple ensembles of feedforward neural networks built on feature-engineered inputs (Racle et al., 2023; Reynisson et al., 2020). In addition, existing works (Reynisson et al., 2020; Jensen et al., 2018; Racle et al., 2023; You et al., 2022; Cheng et al., 2021; Wang et al., 2024) focus sorely on the peptide-level interaction, which overlooks the importance of biological context (e.g., the source antigen) within the MHC-II antigen presentation pathway.

Motivated by these challenges, we curate a high-quality, large-scale dataset for modeling MHC-II antigen presentation in humans across immunological scales, followed by a comprehensive benchmark

study. The experimental peptide samples, initially collected from the Immune Epitope Database (IEDB) (Vita et al., 2018) and other public sources (e.g., (Reynisson et al., 2020; Racle et al., 2023)), undergo rigorous data filtering, data splitting with strict and practical constraints, antigen information alignment, antigen-aware augmentation, and additional data integration from third-party algorithms (e.g., predicted MHC-II structure from AlphaFold3 (Abramson et al., 2024) and estimated binding core via motif deconvolution (Racle et al., 2019b)). This effort not only expands and standardizes the existing peptide-MHC-II datasets, but also introduces a novel antigen-MHC-II dataset that supports the more comprehensive antigen-based modeling and evaluation.

Based on the curated dataset, we employ three major machine learning tasks that capture different stages of the MHC-II antigen presentation pathway: peptide binding affinity (BA) prediction, peptide eluted ligand (EL) presentation prediction, and antigen EL presentation prediction. While the first two tasks are well-established in existing works, our work is the first attempt that address MHC-II presentation at the antigen level, as there exists no antigen datasets or antigen-based methods for this problem. The antigen modeling task reflects a broader biological process within the presentation pathway that peptide-based tasks overlook (i.e., antigen processing stage). After training, a multi-scale evaluation framework is employed to benchmark both the model preciseness and efficiency in identifying epitope candidates. We conduct a comprehensive benchmark analysis using a modular architectural framework that is able to accommodate various modeling designs commonly used in AI for science, including alternative input configurations, model architectures, and training strategies. We also evaluate state-of-the-art peptide-MHC-II models on our dataset to establish strong baseline references. While this dataset is grounded on biological domain knowledge in immunology, it reflects the practical and fundamental challenge of how fine-grained biomolecular interactions can be learned from large-scale sequence data. This challenge underlies many tasks in AI for science, where experimental complex structures are often not accessible.

Our contributions can be summarized as: (1) the curation of a large-scale dataset for human MHC-II antigen presentation, which supports not only the well-established peptide prediction but also the novel task of antigen prediction, (2) the construction of a benchmark task with better MHC-II coverage, peptide diversity, and binding core constraints, (3) the introduction of a multi-scale evaluation framework that assesses model performance across immunological scales, providing deeper insights into model behavior and generalizability, and (4) a benchmark study that offers strong baseline results and valuable insights into modeling design choices to guide future ML developments.

## 2 BACKGROUND AND RELATED WORK

### 2.1 MHC-II ANTIGEN PRESENTATION

The MHC-II antigen presentation pathway typically involves five stages: (1) The uptake of exogenous antigens into antigen-presenting cells (APC), (2) antigen processing into peptide fragments, (3) peptide-MHC-II binding into stable complexes, (4) the presentation of these complexes to the cell surface, and (5) the recognition by $CD4^+$ T-cells, initiating immune responses like cytokine secretion (Pishesha et al., 2022). A high-level illustration of this process is provided in Figure A2.

Three major types of data are considered: binding affinity (BA), assessed using in vitro binding assays, reflects the binding strength between peptides and MHC-II (Stage 3); eluted ligand (EL) presentation, obtained via mass spectrometry (MS) after peptide elution from MHC-II, indicates if peptides are presented on the cell surface (Stages 3∼4); T-cell immune response data reflects the recognition of presented peptides by $CD4^+$ T-cells (Stage 5), which is the most relevant to immune outcomes. These data types are highly correlated along the antigen presentation pathway (Weingarten-Gabbay et al., 2024; Wu et al., 2019) with some subtle differences. For example, weak binders may still elicit T-cell responses if they are stably bound to MHC-II and efficiently presented (James & Kwok, 2008). In this work, we mainly focus on BA and EL data, and further extend EL with antigen information to cover the biological processes from antigen processing to peptide presentation (Stage 2∼4).

### 2.2 PEPTIDE-MHC-II DATASETS

Existing datasets for MHC-II antigen presentation largely come from the Immune Epitope Database (IEDB), which covers experimentally validated peptides from literature and direct submissions. However, its raw data is not directly formatted for ML purpose due to annotation noise, ambiguous labels, and inconsistent experimental approaches. For BA data, NetMHCIIpan3.2 (Jensen et al.,

Table 1: Comparison of train sets used in existing works (Net4.2/3 = NetMHCIIPan4.2/3, RPE = RPEMHC, Mix2 = MixMHC2Pred2). Number of peptide cluster indicates the peptide diversity. Notably, our dataset is the first one that supports antigen-based modeling for MHC-II presentation.

| Train set | $BA_{peptide}$ | | | $EL_{peptide}$ | | | | $EL_{antigen}$ |
|---|---|---|---|---|---|---|---|---|
| | Ours | Net4.2/3 | RPE | Ours | Net4.2 | Net4.3 | Mix2 | Ours |
| *#Pair* | 136K | 126K | 131K | 634K | 123K | 339K | 558K | 46,539 |
| *#MHCII* | 77 | 72 | 72 | 132 | 43 | 56 | 76 | 121 |
| *#Cluster* | 5,698 | 4,998 | 4,942 | 62,461 | 18,508 | 30,424 | 61,432 | 30,709 |

Table 2: Comparison of test sets used for evaluating performance in peptide-MHC-II prediction. "Mixed" indicates that labels are collected from varying experimental measures. "Immune" means the label is taken from reported $CD4^+$ T-cell immune response.

| Test set | Ours | ID2017 | BD2020 | $IC50_{test}$ | $\text{T-cell}_{epitope}$ | $CD4_{epitope}$ | Neodb |
|---|---|---|---|---|---|---|---|
| *#Pair* | 3,867 | 857 | 64,954 | 2,413 | 1,698 | 917 | 128 |
| *#Seq* | 2,608 | 163 | 18,770 | 552 | 1,112 | 713 | 120 |
| *#MHCII* | 80 | 10 | 49 | 47 | 36 | 20 | 36 |
| *#MHCII DR* | 30 | 10 | 49 | 25 | 31 | 20 | 24 |
| *#MHCII DP* | 29 | 0 | 0 | 10 | 1 | 0 | 7 |
| *#MHCII DQ* | 21 | 0 | 0 | 9 | 2 | 0 | 5 |
| *Antigen Info* | ✓ | ✗ | ✗ | ✗ | ✓ | ✓ | ✗ |
| *Strict 9-mer* | ✓ | ✗ | ✗ | ✗ | ✗ | ✓ | ✗ |
| *Label* | BA, EL | BA | Mixed | BA | Immune | Immune | Immune |

2018) curated the widely used IEDB2016 dataset by selecting records with valid IC50 (half maximal inhibitory concentration, a common measure for binding affinity) values from IEDB. It contains 126K human peptide-MHC-II binding pairs. For EL data, NetMHCIIpan series (Reynisson et al., 2020; Nilsson et al., 2023b;a) and MixMHC2pred2 (Racle et al., 2023) each curated their own training data from mass-spectrometry (MS) records in public (e.g., IEDB) and in-house sources. One key issue of EL data compared to BA is its class imbalancing, with most of the documented EL results being positive. This requires negative augmentation for effective model training. NetMHCIIpan series randomly samples negative decoy peptides of the same length from the human proteome, while MixMHC2pred2 samples unobserved peptides from their source antigen as negatives. Even though the latter antigen-aware augmentation better follows biological context, it requires access to antigen information, which is not always available.

The test data, by contrast, is less standardized. Researchers typically construct their own test sets by either extracting non-overlapping entries from IEDB (e.g., ID2017 (You et al., 2022), BD2020 (Venkatesh et al., 2020), $IC50_{test}$ (Cheng et al., 2021), $\text{T-cell}_{epitope}$ (Jensen et al., 2018), $CD4_{epitope}$ (Reynisson et al., 2020)) or generating data via wet-lab experiments (e.g., Neodb (Wu et al., 2023), DFRMLI (Zhang et al., 2011)). One important and widely accepted constraint is to exclude any peptide that contains a 9-mer (9-residue subsequence) previously seen in training, whereas only the construction of $CD4_{epitope}$ strictly follows this criteria. This leads to potential information leakage and overestimated performance for most test sets. On the other hand, MHC-II distribution in most data is highly skewed towards the DR alleles, leaving other MHC-II classes (i.e., DP, DQ) underrepresented. Moreover, antigen information is often absent, making antigen-level evaluation infeasible.

## 2.3 PEPTIDE-MHC-II MODELING

Several machine learning methods were proposed for modeling peptide-MHC-II interaction. The NetMHCIIpan (Nielsen et al., 2008; Jensen et al., 2018) family utilizes the NNAlign (Nielsen & Lund, 2009) framework, which is an ensemble method of feedforward neural networks (FNNs) with feature-engineered inputs of peptide and MHC-II sequence. NetMHCIIpan4 (Reynisson et al., 2020; Nilsson et al., 2023b;a) series further extends this approach using NNAlign_MA (Alvarez et al., 2019) to handle multi-allele data, which is beyond the scope of this paper. MixMHC2pred (Racle et al., 2023; 2019a) adopts a two-stage feature-engineered pipeline that predicts MHC-II binding specificity and peptide presentation sequentially using FNNs. Advanced deep learning methods, on the other hand, are less explored in this domain. Researchers typically use bidirectional LSTM (Venkatesh et al., 2020), 1D convolutional encoder (You et al., 2022), or a pretrained protein BERT model (Cheng et al., 2021) to encode both peptide and MHC-II sequences, followed by attentive

pooling (Venkatesh et al., 2020; Wang et al., 2024), dot-product operation (You et al., 2022), or multi-head cross-attention (Shen et al., 2025) to capture the peptide-MHC-II interaction. In this work, we experiment with various sequence encoders followed by cross-attention module to capture the peptide-MHC-II interaction.

# 3 DATASET DESCRIPTION

Our dataset focuses on the human MHC-II antigen presentation pathway, and is built upon two experimental measures: binding affinity (BA) and MS-based eluted ligand (EL) presentation. In addition to the conventional peptide-MHC-II data, we further extract antigen information from public sources and build a comprehensive dataset for antigen-MHC-II presentation.

## 3.1 DATA COLLECTION

Our dataset integrates public data from multiple sources. For peptide-MHC-II BA data, we take the well curated IEDB2016 (Jensen et al., 2018) and enrich it with BA records from the latest MHC-II ligand assay in IEDB (Vita et al., 2018) (accessed on Feb 16, 2025). After binding pairs de-duplication, we further filter out entries with ambiguous BA labels (e.g., IC50 > 1000nM) and non-human MHC-II. The BA labels are normalized into $[0, 1]$ via the transformation $1 - \log(\text{IC50})/\log(50000)$. After these processing steps, we collect $\sim$141K binding pairs, covering 78 unique human MHC-II.

For peptide-MHC-II EL data, we start by aggregating the compiled MS-based datasets from NetMHCI-Ipan4 (Reynisson et al., 2020) and MixMHC2pred2 (Racle et al., 2023). NetMHCIIpan4 is trained on data from 16 public sources, while MixMHC2pred2 is trained on data from 30 public sources. Both methods also incorporate their in-house datasets. However, these datasets are all at the peptide-level, which only addresses the biological stages following peptide binding. Our goal is to build a multi-scale dataset that can capture a broader scope of antigen presentation pathway. We first enrich the existing samples by incorporating the latest EL records from IEDB, followed by de-duplication, removal of non-human MHC-II and ones with conflicting labels. Eventually, we are able to collect $\sim$1.2M peptide-level EL data, covering 134 unique human MHC-II. To further enable antigen-level training, we extract all the available antigen information from IEDB and perform peptide-antigen alignment. This supports our proposed antigen-level task and evaluation, which further reflects the upstream stage of antigen processing. We successfully assign antigen information to $\sim$219K peptide-MHC-II pairs, covering 10,023 unique antigen sequences. As shown in Table 1, our dataset is more comprehensive, with better MHC-II coverage and peptide diversity compared to existing ones.

## 3.2 DATA SPLITS CONSTRUCTION

The data splits for BA, $\text{EL}_{\text{peptide}}$, and $\text{EL}_{\text{antigen}}$ datasets are carefully constructed, with consideration of MHC-II coverage, antigen information availability, and orthogonality of binding motifs. We also prevent peptide overlap between training and testing across BA and EL tasks to provide an easy setup for joint training, which has been shown to improve performance on individual tasks (Reynisson et al., 2020; Barra et al., 2018).

Candidate test samples for BA and EL are first selected from IEDB using a year cutoff of 2020. To prevent data leakage during joint training, peptides appearing in the other's training set are reassigned to training. Peptides lacking antigen information are also moved to training. In addition, for strict and practical evaluation, common practice (Jensen et al., 2018; Reynisson et al., 2020; Nielsen et al., 2007) argues that no 9-mer (i.e., 9-residue subsequence) in the test peptides should appear in training. We iteratively move peptides from test to training with continuous verification of 9-mer overlaps until convergence. Our final test sets include 938 BA and 2,929 EL peptide-MHC-II pairs, covering 28 and 73 unique MHC-II, respectively. These sets are comparable in size to prior work but offer stricter evaluation, broader MHC-II coverage, and antigen annotations (Table 2).

For validation set of antigen-level tasks, the initial validation samples come from random selection of peptide clusters generated by the CD-HIT algorithm (Fu et al., 2012) instead of a year cutoff. Then, peptides with no antigen information and seen 9-mers are moved to training. This ensures that observed peptides in antigen-level tasks are also out-of-distribution. Note that the same antigen may appear across data splits. It reflects the practical scenario where biologist seeks to explore alternative peptides within the antigen even when known epitope exists. We further expand the peptide-level validation set from peptides without antigen information. We apply stratified sampling based on the MHC-II distribution, while controlling for the peptide overlap ratio. As a result, the final validation

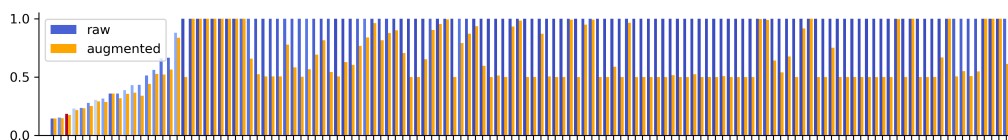

Figure 1: Label positive ratio of peptide-level eluted ligand (EL) data for each MHC-II molecule, before (left) and after (right) the data augmentation and label re-balancing. The red bar highlights the raw label distribution of the DRB10101 MHC-II type (main contributor of negative examples).

sets contain 6,958 and 54,351 peptide-MHC-II pairs for BA and EL data (roughly 5% of training data), respectively. The peptide overlap ratio is controlled at around 25%, with consistent MHC-II distribution between training and validation. Detailed data statistics are included in Table A6.

### 3.3 Label Re-balancing and Data Augmentation

Although the peptide-level EL data appears to be globally balanced (603K positives versus 511K negatives), the label distribution is highly skewed per MHC-II. As shown by the left bar of the side-by-side barplot in Figure 1, 86% of MHC-II molecules are associated with only positive peptides, while DRB10101 (highlighted in red) alone contributing to 17% of negative data. The same issue is observed in prior works as well (Nielsen et al., 2008; Racle et al., 2019a). To address this issue, a common approach is to randomly sample decoy peptides of the same length from the human proteome as negative candidates (Reynisson et al., 2020). However, this may result in easily distinguishable negatives, such that the candidates are too dissimilar to positive ones in terms of interaction patterns and immunological relevance. To enable a finer-grained distinction between positive and negative peptide-MHC-II interactions, we extract the neighboring peptides from the same source antigen as the negative samples. These peptides share similar biological context with the positives, and are likely to be processed through endosome but not selected for presentation due to subtle differences in binding motifs. We further use the estimated binding cores from MoDec (Racle et al., 2019b) as guidance, such that the negatives are allowed to have overlaps with the positives without violating the binding cores. We generate four negative augmentations for each positive peptide. Additionally, to further enhance training robustness of sequence-based model, we allow random extension of the peptide at both end and random shifting of peptide window by 1 based on its source antigen. The updated label distribution for each MHC-II is shown by the right orange bar in Figure 1. The persistent label imbalance for some MHC-II samples happens due to the lack of antigen information of their corresponding peptides. As discussed later, we further use an auxiliary task of binding core estimation for improved learning in these MHC-II samples. We also demonstrate that the false negative rate of our augmentation approach is almost negligible in Appendix.

### 3.4 Additional Data Enrichment

In addition to the data collection, we compute and annotate multiple items that could potentially enhance model learning for both input features and output labels. We first extract the residue-level ESM2 (Lin et al., 2023) embedding as the additional sequence feature. It is one of the most widely used protein language models that has shown to have implicit structural knowledge. In addition, we estimate the binding motifs within each positive peptide from motif deconvolution using MoDec (Racle et al., 2019b). As we will show later, it can serves as a pseudo-label for the auxiliary task of binding core prediction. We further infer the MHC-II structures via AlphaFold3 (Abramson et al., 2024) to include explicit structural information as input. We avoid computing the peptide structures from two perspectives. Biologically, unbound peptide conformations often differ from their bound states within the MHC-II complexes, which makes the predicted peptide structures unlikely to reflect the true conformation in complexes (Ayres et al., 2017). MHC-II, on the other hand, has a relatively rigid binding groove and stable conformation. Computationally, it is also infeasible to compute millions of structures of diverse peptides in both training and inference. Detailed descriptions of MoDec and AlphaFold3 are included Appendix, as well as the quality analysis of MHC-II predicted structures and the sensitivity analysis of models' outputs towards structural noise.

### 4 Benchmark Tasks and Evaluation

The curated dataset enables various machine learning tasks that align with different stages of the antigen MHC-II presentation pathway. Besides the well-established tasks of peptide binding affinity

(BA) and eluted ligand presentation (EL) prediction, we introduce a novel antigen-level EL task that aims at identifying immunologically important regions within full antigen sequences. To better evaluate the model performance, we employ a multi-scale evaluation framework, incorporating both standard peptide-level and epitope-level metrics, and a novel antigen-level coverage-redundancy analysis. Table A7 provides an overview of the mapping between evaluation methods and benchmark tasks.

## 4.1 BENCHMARK TASKS

At the peptide level, the model predicts (1) BA between peptides and MHC-II as a regression task, and (2) EL presentation by the given MHC-II as a binary classification task. However, one of the issues with peptide-based modeling is the absence of antigen context. From data analysis, we observe that the same peptide can have contradictory labels across different antigens. For example, the CD4 epitope benchmark (Jensen et al., 2018) contains 35 out of 713 peptides that have opposite labels. This may arise from factors like variations in antigen processing or competition among neighboring peptides in the biological processes.

To address this issue, we further introduce the third task of (3) antigen-level EL presentation. Given an antigen sequence and an MHC-II, the goal is to identify regions of immunological importance (i.e., predict the likelihood of each amino acid being positive). This task goes beyond peptide modeling and requires the model to reason over the full antigen sequence as a richer biological context. Performance on this task reflects a model's ability to capture three stages in presentation pathway, including antigen processing, peptide binding, and peptide presentation. The corresponding evaluation method is described below.

## 4.2 MULTI-SCALE EVALUATION ACROSS IMMUNOLOGICAL SCALES

To examine both the accuracy and the efficiency of the model in identifying epitope candidates to MHC-II presentation, we employ a multi-scale evaluation framework. In addition to the peptide-level and epitope-level metrics used in prior studies, we introduce a novel antigen-level evaluation method that enables a global and fine-grained view of model performance across antigen sequences.

**Peptide-level Evaluation:** As the most straightforward way of evaluating peptide-based model performance, peptide-level metrics directly compare the observed peptide labels from experiments with their corresponding predicted scores. For binding affinity prediction, root mean square error (RMSE) is reported. We also follow the existing works (Jensen et al., 2018; You et al., 2022; Wang et al., 2024) and binarize the binding affinity label IC50 using the threshold of 500nM, a common threshold used to differentiate binders from non-binders, and report the ROC-AUC score. This measures the model's ability in ranking binders higher than non-binders. For eluted ligand classification, we report only the accuracy as the success rate since the test set only contains experimentally verified presented peptides.

**Epitope-level Evaluation:** Epitope-level evaluation examines the model effectiveness in identifying the known epitope from its source antigen. It not only considers the predicted score of the observed peptides, but also the prediction of other unobserved peptides within the antigen, which provides a broader view of model performance. For peptide-based models, evaluation is done by first identifying the source antigen of the epitope. Then, all candidate peptides of the same length as the epitope are generated from the antigen, and predictions are made for each peptide-MHC-II pair. Conventional metrics that fall into this category are *FRANK* score (Reynisson et al., 2020; Jensen et al., 2018; Wang et al., 2024) and $AUC_{epitope}$ (Wang et al., 2024) score. FRANK computes the fraction of peptides with a higher predicted scores than the known epitope. In other words, it measures the false positive rate. The $AUC_{epitope}$ score is measured by assigning negative labels to all peptide candidates other than the epitope and report the ROC-AUC score. In this work, we directly adapt these metrics to our BA and EL test data. While the positive peptides in both data are not strictly validated epitope, we adopt the term "epitope" for convenience purposes.

Even though epitope-level evaluation is more comprehensive than the direct peptide-level evaluation, several limitations remain. One of the biggest issues is that it overlooks cases where multiple epitopes exist within a single antigen. As a result, the same peptide may be treated inconsistently as positive and negative across evaluation rounds. For example, 77 out of 140 antigen in the CD4 epitope benchmark (Reynisson et al., 2020) contain multiple epitopes, leading to 653 out of 713 unique peptides being inconsistently labeled at least once. Considering multiple epitopes can be indeed

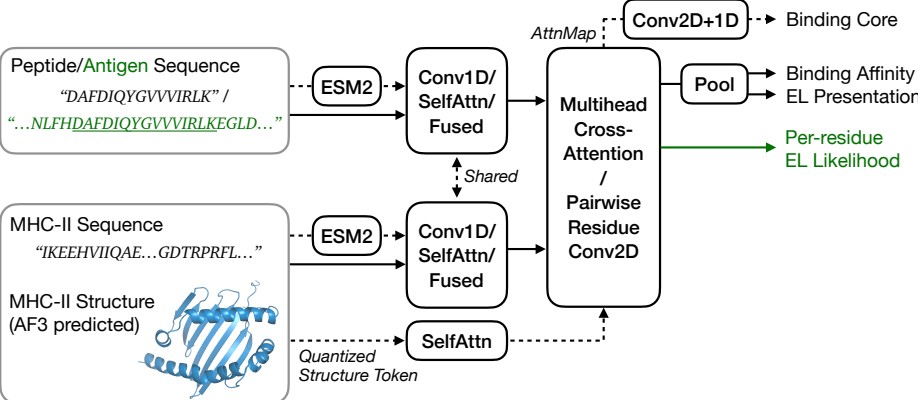

Figure 2: Overview of the architectural framework used in our benchmark study. The dashed lines indicate optional settings used for ablation analysis, which includes the use of ESM2 embeddings, structural features, and binding core prediction auxiliary task. Antigen modeling is shown in green.

challenging when evaluating peptide-based models, especially when positive peptides have varying length. For starter, highly overlapped and redundant peptide candidates need to be generated from the given antigen, which drastically increases the computational complexity. Meanwhile, as the number of epitopes increases, the computed metric becomes less comparable across antigen as the amount of negative candidate also scales linearly with respect to the length of antigen.

**Antigen-level Evaluation:** Inspired by object detection metrics, we propose an antigen-level evaluation that examines the tradeoff between region-level coverage and redundancy in the predicted EL regions for each antigen. This provides a global and fine-grained view of the model's ability to capture epitope candidates from antigen, while mitigating the limitations of epitope-level evaluation.

We first compute the per-residue labels as the count of inclusion from epitopes identified from experiments: for each residue $r_i$, $\text{label}_i = \sum_j \mathbb{1}(r_i \in E_j)$ , where $E_j$ is the $j$th epitope within the antigen and $\mathbb{1}(...)$ being the indicator function. We then define ground truth regions $G = \{G_1, ..., G_n\}$ as $n$ contiguous non-overlapping segments of residues where the label is nonzero. Note that $n \leq |E|$ since overlapped epitopes are aggregated into one region. The per-residue prediction can be extracted intuitively from antigen-based models, and can be approximated by aggregating the predicted scores along a fix-length sliding window from peptide-based models. We set the length to be 9, which is the conventional size of binding cores. The $m$ predicted regions $P = \{P_1, ..., P_m\}$ is then defined similarly as contiguous segments with the score passes a given threshold. Based on the two region sets $G$ and $P$, we compute region-level coverage and redundancy as follow:

*Region-level Coverage* measures how well the predicted regions cover the ground truth regions, computed by the weighted sum of residue overlapped ratio between $G_i$ and $P$.

$$\text{Coverage} = \sum_{i=1}^{n} \tilde{w}_i \left( \frac{\sum_j (|G_i \cap P_j|)}{|G_i|} \right) \tag{1}$$

where $\sum_j (|G_i \cap P_j|)$ simply represents the total number of residues in $G_i$ predicted as positive, and weight $\tilde{w}_i$ is the sum of log scale of residue-level labels within the ground truth region $G_i$, normalized by both the region size and the total number of regions across antigen. The log scaling retains the ranking of region importance, while compressing the label magnitude to be more reasonable.

$$\tilde{w}_i = \frac{w_i}{\sum_{j=1}^{n} w_j}, \ \ w_i = \frac{1}{|G_i|} \sum_{r_j \in G_i} \log(1 + \text{label}_j) \tag{2}$$

*Redundancy* computes the number of residues within all predicted regions normalized over the length of antigen $\mathcal{A}$, which indicates the opposite of prediction sparsity.

$$\text{Redundancy} = \frac{1}{|\mathcal{A}|} \sum_i \sum_j \mathbb{1}(r_j \in P_i) \tag{3}$$

Table 3: Comparison of different input configurations and training strategies evaluated on peptide BA and EL tasks. The full results are included in Appendix.

| Input | | Strategy | | Binding Affinity | | Eluted Ligand | | |
|---|---|---|---|---|---|---|---|---|
| ESM2 | Struct | Joint | Aux | AUC | $AUC_{epitope}$ | Accuracy | $AUC_{epitope}$ | CR-AUC |
| ✓ | | | | 0.7547 | 0.7717 | 0.6470 | 0.8253 | 0.6048 |
| | | | | 0.7313 | 0.7615 | 0.6098 | 0.8095 | 0.6101 |
| ✓ | ✓ | | | 0.7367 | 0.7564 | 0.6582 | 0.8264 | 0.6198 |
| ✓ | | ✓ | | 0.7473 | 0.7747 | 0.6554 | 0.8328 | 0.6045 |
| ✓ | ✓ | ✓ | | **0.7656** | 0.7658 | 0.6763 | 0.8372 | 0.6420 |
| ✓ | ✓ | ✓ | ✓ | 0.7627 | **0.8127** | **0.6955** | **0.8492** | **0.6634** |

Table 4: Performance comparison of existing peptide-based models. The asterisk (*) indicates that the test data is filtered with valid inputs under MixMHC2Pred2's constraints for fair comparison. Ours<model> represents our replicate of existing models. The full results are included in Appendix.

| Method | Binding Affinity | | Eluted Ligand* | | |
|---|---|---|---|---|---|
| | AUC | $AUC_{epitope}$ | Accuracy | $AUC_{epitope}$ | CR-AUC |
| NetMHCIIPan4.3 (Nilsson et al., 2023a) | **0.8115** | 0.8236 | 0.4980 | **0.8672** | 0.6526 |
| NetMHCIIPan4.3context | 0.7627 | 0.8160 | 0.5314 | 0.8646 | 0.6510 |
| RPEMHC (Wang et al., 2024) | 0.7978 | **0.8436** | - | - | - |
| MixMHC2Pred2 (Racle et al., 2023) | - | - | 0.3462 | 0.8658 | 0.6906 |
| ImmuScope (Shen et al., 2025) | - | - | 0.6570 | 0.8549 | 0.6796 |
| OursRPEMHC | 0.7713 | 0.7978 | 0.6993 | 0.8642 | 0.7210 |
| OursImmuScope | 0.7927 | 0.8227 | 0.7162 | 0.8601 | 0.7175 |
| Ours (best from Table 3) | 0.7627 | 0.8127 | **0.7347** | 0.8662 | **0.7349** |

We then evaluate the tradeoff between coverage and redundancy, and report the *Coverage-redundancy Area Under the Curve (CR-AUC)* score. In general, both coverage and redundancy tend to increase monotonically as the threshold gets stricter (i.e., increases from 0 to 1). A coverage-redundancy curve can be constructed by varying the threshold used for defining the predicted regions. It captures the model efficiency in capturing biologically meaningful regions. A steep initial rise (Figure A3a) indicates that confident predictions are sufficient to localize ground truth regions. Meanwhile, a shallow or flattened curve (Figure A3c) shows less effective prediction, where additional region proposal fails to substantially improve the coverage. We then report the CR-AUC score. A higher value reflects a more favorable tradeoff, archiving high coverage with low redundancy. Based on the normalization above, CR-AUC lies within $[0, 1]$, and is comparable across models and antigens.

## 5 EXPERIMENTS

We employ various experimental settings to benchmark our curated datasets, including different task formulations, input features, and training strategies. We also compare the BA performance with RPEMHC (Wang et al., 2024) and NetMHCIIPan4.3 (Nilsson et al., 2023a), and EL performance with NetMHCIIPan4.3, MixMHC2Pred2 (Racle et al., 2023), and ImmuScope (Shen et al., 2025), which represent the latest methods in this domain. We also include results from NetMHCIIpan-4.3 using its context-encoding option, which allows the model to use three neighboring residues on each side of the peptide as additional context for prediction. In addition, we built our own modular experimental framework to provide insights behind different modeling choices (e.g., input configuration, model architectures, training strategies) to this problem (Figure 2). Our best model architecture uses a fused module to encode peptide/antigen sequences, a self-attention module to encode MHC-II sequence and structure, and a multi-head cross-attention module to capture the biological interactions. Full model details are described in Appendix C.1. Following the evaluation protocol in Table A7, we report the performance on both peptide binding, peptide presentation, and antigen presentation tasks. Overall, the model results establish strong baselines and modeling insights for both the peptide and antigen tasks, providing useful reference points for future ML work.

### 5.1 EXPERIMENTAL RESULTS ON PEPTIDE BINDING AND PRESENTATION

**Input Configuration:** Three types of features are controlled in our experiments. Following the work in (Koh et al., 2024), we leverage (1) physicochemical residue-level features to initialize the residue

Table 5: Antigen EL performance of peptide- and antigen-based model. The asterisk (*) denotes the same test data setup as in Table 4, making their CR-AUC scores comparable. The rest of the CR-AUC scores are comparable with the results in Table 3.

| Method | Peptide-based | Antigen-based | | | | | | |
|--------|---------------|--------|--------|--------|--------|--------|--------|---------|
| $k$ | - | 32 | 64 | 128 | 512 | 1024 | random | random* |
| CR-AUC | 0.6092 | 0.6346 | 0.6409 | 0.6463 | 0.6402 | 0.6340 | **0.6649** | 0.6808 |

embedding. We further consider the usage of (2) ESM2 (Lin et al., 2023) protein language embedding of both peptides and MHCII for its implicit knowledge of protein structure, and (3) the predicted MHC-II structures from AlphaFold3 (Abramson et al., 2024) as the additional structural inputs. As shown in Table 3, performance drops significantly for both tasks without ESM2 embedding. The incorporation of MHC-II structural information significantly improves over settings without structural inputs in EL task, while the results in BA task show mixed patterns. This could be attributed to the greater amount of data required to effectively capture the sequence–structure relationship, while BA data is about 10 times fewer than EL data.

**Training Strategy:** We further evaluate how training strategies affect the performance. We first examine the effects of joint training on BA and EL performance. As shown by the first and forth row of Table 3, joint training has shown to have improvement in some metrics. We then examine the effect of auxiliary supervision on peptide binding core prediction. The binding core is predicted using a 2D convolution over the cross-attention map between peptides and MHC-II. Given that attention maps have the potential of capturing spatial proximity between residues (Lin et al., 2023), we hypothesize that the attention map alone can infer the binding core largely determined by spatial interaction. As shown in the last row of Table 3, the auxiliary task significantly improves the performance of both tasks. Despite label re-balancing and data augmentation, some MHC-II can still have extremely skewed label distribution (Figure 1). The auxiliary core prediction tasks allows the model to localize meaning patterns from peptide-MHC-II interaction, even in cases where all associated labels are positive. We argue this as the main reason for the observed performance improvement.

**Method Comparison:** We further compare our model using the best configuration above with existing methods in Table 4. All performance results are obtained from the publicly released models. NetMHCIIpan4.3 and MixMHC2Pred2 only provide precompiled models with limited implementation details. To examine how key architectural differences in RPEMHC and ImmuScope may affect performance, we additionally train two model variants that replicate their design choices. Ours$_{RPEMHC}$ replaces the peptide-MHCII cross-attention module with a 2D convolution over pairwise residue features, while Ours$_{ImmuScope}$ augments our model with additional convolutional refinement blocks for peptide representations after cross-attention. For BA task, our model performs slightly worse than RPEMHC and NetMHCIIPan4.3. One of the reasons might be related to checkpoint selection. Currently, the best checkpoint of the joint BA-EL training is chosen based on the average peptide AUC scores across both tasks. However, since BA has much less data compared to EL, the selected checkpoint could be biased towards EL performance. For EL task, we first filter our test set according to MixMHC2Pred2's input constraints (i.e., peptides composed of natural amino acids with lengths 12-21) for a fair comparison across models, which reduces the test size from 2929 to 2484. We also increase the sliding window size from 9 to 12 in antigen-level evaluation. In general, our approach shows stronger performance, especially on peptide-level and antigen-level metrics. We realized that MixMHC2Pred2 is relatively more conservative in its scoring. Its highest peptide score averaged across all test antigen is 0.438, while NetMHCIIPan4.3 is 0.572. This could explain its low peptide accuracy measured by the probability threshold of 0.5. Our best model performs slightly better than Ours$_{RPEMHC}$ on average, which is expected since 2D convolution is less efficient than cross-attention at capturing global interactions. Ours$_{ImmuScope}$ shows improvement on BA specifically. Since BA signals are more sensitive to the binding core, the additional convolutional refinement may help the model focus on the most relevant local regions for prediction.

## 5.2 EXPERIMENTAL RESULTS ON ANTIGEN PRESENTATION

The antigen-based model shares the same model architecture as the peptide-based model, except the prediction head is modified into a position-wise (residue-level) prediction layer without global pooling. The main objective of the antigen modeling task is to identify the antigenic regions most relevant for MHC-II presentation, which is evaluated using the proposed CR-AUC score. In addition,

for efficient training, antigen sequences are truncated to a maximum window size $k$ to avoid CUDA out-of-memory errors. Instead of sampling arbitrary subsequences of length $k$, we only sample from "valid" regions, where no known epitope is being cut through. This preserves biologically meaningful regions for training. The evaluation is performed on the full antigen sequence without any truncation. We then compare the performance of antigen-based models with varying $k$ with the performance of the peptide-based models trained only on the peptide EL task for fairer comparison (Table A2).

As shown in Table 5, the best-performing antigen-based model significantly outperforms the peptide-based model by a large margin. Notably, the antigen-based models have only seen around 25% of positive peptides available for training peptide-based models, which further highlights the promising potential of antigen-based modeling in solving the antigen EL task. The choice of window size $k$ also influences the performance. As $k$ decreases to small values (e.g., from 128 to 32), the antigen modeling will gradually reduce to peptide modeling, which results in performance drop. On the other hand, although increasing $k$ (e.g., from 128 to 1024) will provides richer biological context, the training difficulty also increases as the residue-level label distribution becomes less balanced. Instead of hand-picking a fixed window size to balance the trade-off, we propose the randomized window sizing, where $k$ is sampled at each iteration from a predefined set instead of being fixed. We use the set {64, 128, 256, 512, 1024} in our experiment, which corresponds to the result of "random". It reaches the best performance with a CR-AUC score of 0.6649. "random*" corresponds to the same setup in Table 4, where the test data is filtered according to MixMHC2Pred2's constraints. Therefore, its CR-AUC value is directly comparable to the CR-AUC reported in Table 4. This result outperforms almost all existing baselines, but falls slightly behind our best peptide-based models that uses joint training and the core-prediction auxiliary task. A promising direction for improving future antigen-based models is to incorporate more diverse supervision signals at training (e.g., predicting whether an epitope exist within antigenic regions as a global label), which we leave for future work. A qualitative analysis of the coverage-redundancy curve for the best peptide-based and antigen-based models is provided in Appendix D.5, which further highlights the potential of antigen-based models in localizing candidate epitopes with high confidence.

## 6 DISCUSSION

We curate a comprehensive and large-scale dataset for human MHC-II antigen presentation prediction. It supports three major ML tasks, including a novel antigen-level task that captures broader biological processes within the presentation pathway. We further employ a multi-scale evaluation framework to comprehensively analyze the model performance. Via extensive experiments, we find that joint training, structural inputs, and auxiliary binding core prediction can improve performance on both peptide BA and EL tasks. Meanwhile, antigen-based modeling, which incorporates richer biological context, has shown its great potential in localizing epitope candidates within antigen sequence.

For future work, we plan to expand the structural component of our dataset using the peptide-MHC-II complex structures via AlphaFold3. The co-folding model is expected to have a better implicit knowledge of inter-chain residue interactions, which will be reflected in its predicted complex structures. It is also promising to explore other advanced approaches (e.g., constructing protein graphs from estimated contact map (Koh et al., 2024), or directly applying equivariant models (Fuchs et al., 2020; Satorras et al., 2022) to encode protein geometry) to further improve performance in antigen presentation.

## 7 LIMITATION

One limitation of this work is that both BA and EL labels are indirect proxies for T-cell immune responses. While they provide useful signals for epitope likelihood, they do not fully capture downstream immunogenicity. Unfortunately, T-cell response data remains too scarce to support large-scale training. In addition, antigen annotations are missing for a subset of peptides, which may introduce selection bias in the subset used for training antigen-level models. Another limitation is that our study focuses exclusively on single-allele data, where the peptide-MHC-II mapping is certain. In contrast, real-world MS data often involves multi-allele samples, where a positive label only indicates that at least one MHC-II within a group is responsible for the peptide presentation. Extending our framework to incorporate multi-allele data is an important direction for future work, and may benefit from strategies like multi-instance learning (Alvarez et al., 2019; Ilse et al., 2018).

## 8 ETHICS STATEMENT

This work does not raise any ethical concerns.

## 9 REPRODUCIBILITY STATEMENT

The data collection and processing steps are detailed in Section 3. The implementation details, model specification, and training hyperparameters are comprehensively discussed in Appendix C. Upon acceptance, the curated dataset will be released, as well as the code repository for the multi-scale evaluation and our experimental pipeline.

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

# A    BACKGROUND OF MHC-II ANTIGEN PRESENTATION PATHWAY

MHC-II proteins are a class of major histocompatibility complex molecules primarily present antigenic epitopes on the surface of antigen-presenting cells (APCs). They are encoded by genes in the HLA-DP, HLA-DQ, and HLA-DR loci and consist of two chains/domains ($\alpha$ and $\beta$) that together form an open-ended binding groove (Figure A1). This structure allows MHC-II to accommodate peptides of varying lengths. Among the HLA Class II loci, HLA-DR is the most extensively studied, with more available epitope sequence data in public databases. This is attributed to its higher expression level and polymorphism in the human population, which make it more accessible for experimental isolation and characterization.

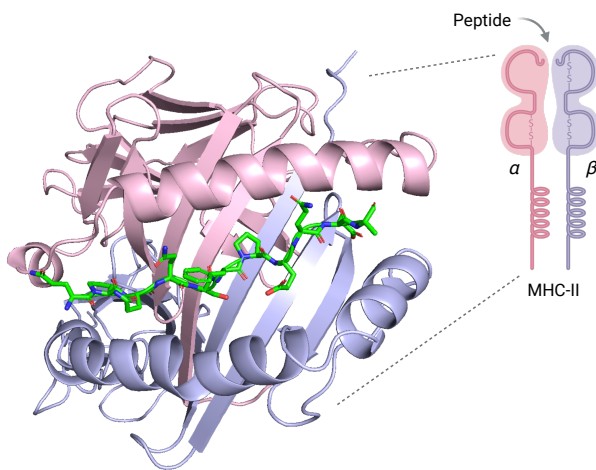

Figure A1: A example visualization of the peptide-MHC-II complex. MHC-II protein contains two chains, with $\alpha 1$ domain colored in pink and $\beta 1$ domain colored in purple. The peptide, colored in green, is bound into the middle part. The open-ended binding groove of MHC-II is formed by two $\alpha$-helices and one $\beta$-sheet.

The MHC-II antigen presentation pathway, as shown in Figure A2, mainly consists of five stages: (1) The antigen-presenting cell (APC) first takes in the antigen. (2) The antigen is then processed and broken down into peptide fragments within the endosomal compartments. (3) MHC-II molecules selectively bind to a peptide and form peptide-MHC-II complexes. (4) The peptide-MHC-II complexes are then transported to the cell surface for presentation. As last, (5) CD4$^+$ T-cells scan the surface of APC and triggers T-cell immune response if the presented peptide is recognized. In our machine learning formulation, peptide binding affinity prediction corresponds to stage (3); peptide eluted ligand prediction captures both stages (3) and (4); while antigen eluted ligand prediction covers stages (2), (3), and (4).

# B    MORE DATA ANALYSIS

## B.1    FALSE NEGATIVE FROM DATA AUGMENTATION

In our experiments, we utilize the antigen-aware augmentation to increase the number of negative peptides given MHC-II. Here, we perform a statistical analysis in the potential false negative rate introduced by this approach. Although it is challenging to precisely quantify the exact ratio in practical web-lab settings, we approximate the ratio by first examining the positions of all positive peptides from the same antigen in our training and validation sets, and then computing the ratio of two peptides being neighbors. We define neighbors as peptides whose starting positions are less than 15 amino acids apart, which is the typical peptide length. Only 5.64% of the positive peptide pairs meet this criteria, indicating that neighboring peptides of a positive peptide are rarely also positive. This suggests that false negatives introduced through augmentation are likely negligible.

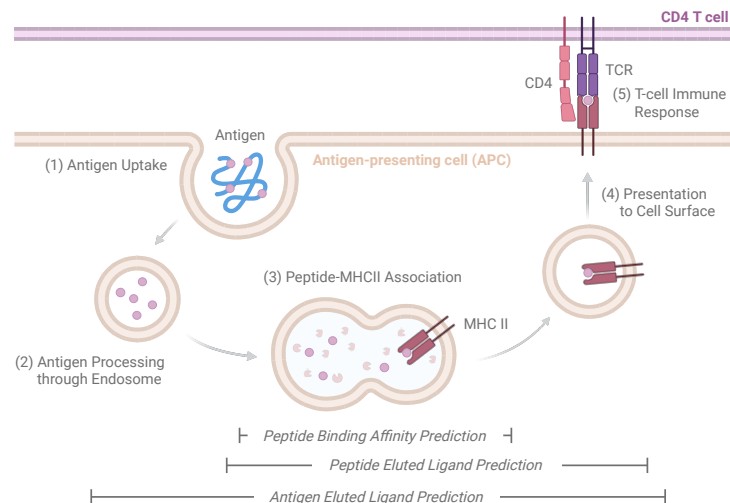

Figure A2: A simplified and high-level illustration of the MHC-II antigen presentation pathway. The process can be broken down into five stages of (1) Antigen uptake, (2) Antigen processing (3) Peptide-MHC-II binding (4) MHC-II presentation of peptide on cell surface, and (5) T-cell immune response.

## B.2  QUALITY OF MHC-II PREDICTED STRUCTURES FROM ALPHAFOLD3

In our experiments, we generate five seeded structures for each MHC-II using AlphaFold3 (AF3), and the one with the highest ranking score (default confidence score provided by AF3) is selected as the final MHC-II structure. Conventional confidence metrics are reported in Table A1, including Predicted TM-score (pTM), Inter-chain Predicted TM-score (ipTM), and Predicted Local Distance Difference Test (pLDDT). According to AF3, the predicted structures are viewed as high-quality for ipTM > 0.8 and pTM > 0.5. The prediction is considered as confident for 70 < pLDDT < 90. To further evaluate the quality of predicted MHC-II structures, we compute the root-mean-square-deviation (RMSD) between predicted structures and experimentally derived structures available on 16 unique MHC-II subtypes. All MHC-II pairs have RMSD < 2.0Å, indicating the predicted structures are highly similar to the experimental structures. A sensitivity analysis of the model's outputs with respect to the structural noise is included in Appendix D.4.

# C  IMPLEMENTATION DETAILS

## C.1  MODEL ARCHITECTURE

As shown in Figure 2, the general model architecture used in this work follows the workflow of encoding, interaction, and prediction. For sequence-based encoding of peptide/antigen/MHC-II, we experiment with self-attention (Vaswani et al., 2017), 1D convolution, and a fused encoder module where 1D convolution and self-attention layers are alternatively applied. ESM2 embedding (Lin et al., 2023), if used, is summed with the residue-level features after a linear projection. For structural input, we discretize the 3D coordinates into structure tokens using ProSST (Li et al., 2024) and encode them with a separate self-attention module.

The interaction module iteratively updates the representations of both peptide/antigen sequence and MHC-II sequence. We experiment with both the multi-head cross attention and the 2D convolution over residue pairwise representations. The latter approach is similar to the RPEMHC (Wang et al., 2024). In settings with structural inputs, the cross-attention updates are performed sequentially from the MHC-II sequence representation to the peptide representation, followed by updates from the MHC-II structural representation to the peptide representation. In our experiments, the MHC-II sequence and structural representations are not updated based on each other. After the interaction updates, attentive pooling is applied, followed by task-specific prediction heads.

Table A1: Quality of MHC-II structures from AF3 measured by confidence metrics and RMSD.

| MHC-II | #Subtype | pTM | ipTM | pLDDT | RMSD (Å) |
|---|---|---|---|---|---|
| DR | 51 | $0.876 \pm 0.020$ | $0.869 \pm 0.021$ | $88.85 \pm 2.03$ | 0.512 |
| DP | 33 | $0.838 \pm 0.041$ | $0.832 \pm 0.055$ | $85.31 \pm 3.87$ | 1.189 |
| DQ | 64 | $0.841 \pm 0.052$ | $0.830 \pm 0.053$ | $85.84 \pm 5.53$ | 0.744 |

For peptide binding affinity and peptide eluted ligand prediction tasks, we apply a bilinear projection layer to integrate the pooled representations of the peptide and MHC-II for final prediction. In contrast, for antigen eluted ligand prediction, no pooling is used after the cross-attention. Instead, a position-wise prediction head is employed to produce residue-level scores. The auxiliary task of binding core prediction is implemented by encoding the cross-attention map between the peptide and MHC-II using 2D convolution, followed by a sliding-window-based (1D convolution) prediction head constructed with 1D convolution. The size of sliding window is set to 9, which corresponds to the conventional size of the binding core.

## C.2 THIRD-PARTY MODEL SPECIFICATION

We use AlphaFold3 (AF3) to generate MHC-II structures. For each MHC-II, we first generate 5 candidate structures via AF3 using model seed 12345 and its default settings (dialect = alphafold3, version = 1). We then choose the structure with the highest default confidence score provided by AF3. For ESM2 embedding, we use the `esm2_t33_650M_UR50D` checkpoint of ESM2 to generate protein language embeddings. Each amino acid will receive a pretrained representation of dimension 1280. For motif deconvolution, we use the MoDec algorithm that finds the motifs and corresponding binding cores given a list of peptides. We used the published version of MoDec-1.2, and ran with the settings: Kmax = 6, L = 9, nruns = 20, mode = MHC2.

## C.3 TRAINING HYPERPARAMETER

All experiments are conducted using the same set of training hyperparameters. Specifically, we use a learning rate of 0.0005 with a total of 50 training epochs, and adopt a cosine annealing scheduler with 10% of the epochs for learning rate warmup. The model is configured with a hidden dimension of 256 and an output dimension of 128 for the final prediction head. A dropout rate of 0.1 is applied throughout each module, except for the final prediction head, where the dropout equals 0.3. Each encoder consists of 4 encoder layers. For self-attention, the number of heads is set to 4. Additional, we employ the multi-kernal 1D convlution with kernel sizes of [5, 9]. For loss computation, we use binary cross-entropy loss for both peptide EL prediction and antigen EL prediction, and mean squared error (MSE) loss for peptide BA prediction. The auxiliary task of binding core prediction is also supervised using binary cross-entropy loss. However, this auxiliary loss is weighted by a factor of 0.1, as it serves primarily as a regularization term and relies on estimated labels.

## C.4 BALANCED SAMPLING DURING TRAINING

To address the label imbalance in the EL data, we employ a balanced sampling strategy during training besides data augmentation. For peptides with positive labels, we randomly sample augmented peptides with a 0.5 probability from either the positive or negative augmentation set at each training step. Note that augmentations are only available for peptides that have been experimentally verified as positive. The antigen EL task follows a similar procedure. At each training step, valid subsequences from antigen truncation are grouped into positive and negative groups. A subsequence is labeled as positive if it contains at least one known epitope. We then randomly sample subsequences randomly from either group to ensure balanced training.

Table A2: Comparison of different encoder choices. The asterisk (*) indicates the setting where peptide and MHC-II share a unified sequence encoder. The full results table is included in Appendix.

| Encoder | | Binding Affinity | | Eluted Ligand | | |
|---|---|---|---|---|---|---|
| Peptide | MHCII | AUC | $AUC_{epitope}$ | Accuracy | $AUC_{epitope}$ | CR-AUC |
| conv | conv | 0.7318 | 0.7389 | 0.6309 | 0.8075 | 0.6014 |
| | conv* | 0.7185 | 0.7165 | 0.6374 | 0.8087 | 0.6016 |
| | self-attn | 0.7288 | 0.7573 | 0.6145 | 0.8084 | 0.6025 |
| | fused | 0.7260 | 0.7437 | 0.6145 | 0.8112 | 0.6085 |
| self-attn | conv | 0.7134 | 0.7270 | 0.6480 | 0.8291 | 0.5882 |
| | self-attn | 0.7330 | 0.7717 | 0.6507 | 0.8328 | 0.5783 |
| | self-attn* | 0.7044 | 0.7491 | **0.6582** | 0.8342 | 0.5915 |
| | fused | 0.7242 | 0.7318 | 0.6514 | **0.8418** | 0.5973 |
| fused | conv | 0.7382 | 0.7574 | 0.6309 | 0.8266 | 0.5921 |
| | self-attn | **0.7547** | **0.7718** | 0.6470 | 0.8253 | 0.6048 |
| | fused | 0.7212 | 0.7474 | 0.6504 | 0.8248 | 0.6058 |
| | fused* | 0.7543 | 0.7594 | 0.6555 | 0.8351 | **0.6092** |

Table A3: Performance difference based on different training data scale.

| Scale | Binding Affinity | | Eluted Ligand | | |
|---|---|---|---|---|---|
| | AUC | $AUC_{epitope}$ | Accuracy | $AUC_{epitope}$ | CR-AUC |
| 100% | 0.7627 | 0.8127 | 0.6955 | 0.8492 | 0.6634 |
| 70% | 0.7584 | 0.7923 | 0.6731 | 0.8382 | 0.6392 |
| 50% | 0.7415 | 0.8051 | 0.6627 | 0.8390 | 0.5978 |
| 30% | 0.7310 | 0.7989 | 0.6412 | 0.8257 | 0.5821 |

# D  MORE EXPERIMENTAL RESULTS

## D.1  ABLATION IN MODEL ARCHITECTURES

We first conduct ablation experiments on model architectures, following the general framework of encoding, interaction, and prediction. For sequence-based encoding of peptide/antigen/MHC-II, we examine self-attention (Vaswani et al., 2017), 1D convolution, and a fused module that alternates between them. The interaction module is built from cross-attention layers to captures the peptides-MHC-II interaction. Then, task-specific prediction heads are applied. The models are trained separately on peptide BA and EL tasks with inputs initialized by residue-level features and ESM2 embeddings. Augmentation is applied for EL tasks. As shown in Table A2, BA performance is much better when peptides are encoded via the fused encoder. This can be attributed to the combination of 1D convolution, which captures the binding core more efficiently, and the self-attention layer, which captures global dependencies. For EL, self-attention encoders generally perform better. One possible explanation is that self-attention, based on its higher expressivity, benefits more from the larger-scale EL data. Based on these results, we use the fused encoder for peptides and self-attention for MHC-II in all other experiments in this work.

## D.2  ABLATION IN DIFFERENT DATA SCALES

In this experiment, we perform an ablation study with respect to data scales to demonstrate the advantages of our curated dataset. Table 1 already shows that our dataset has more data points compared to existing ones, along with better MHC-II coverage and peptide diversity. To quantitatively evaluate how data scale affects the model performance, we re-train our best model using 70%, 50%, and 30% of the training data from random sampling. As shown in Table A3, both peptide-level and antigen-level metrics show large performance improvement as data scale increases. The epitope-level measures, on the other hand, show marginal improvement. As we noted in the main text, epitope-level evaluation can be noisy, less efficient, and biased toward antigens with more verified epitopes. We argue that this is one of the reasons behind the marginal improvement.

Table A4: Performance across different MHC-II alleles (DQ, DP, DR) of our best model.

| MHC-II Type | Binding Affinity | | Eluted Ligand | | |
|---|---|---|---|---|---|
| | AUC | $AUC_{epitope}$ | Accuracy | $AUC_{epitope}$ | CR-AUC |
| DP | 1.0 | 1.0 | 0.7213 | 0.8906 | 0.6883 |
| DQ | 0.7008 | 0.8420 | 0.5898 | 0.8126 | 0.6231 |
| DR | 0.7641 | 0.8079 | 0.7237 | 0.8038 | 0.6386 |

Table A5: Sensitivity analysis of the model outputs with different levels of structural noise.

| Method | Task | $\sigma = 0.1$ | $\sigma = 0.3$ | $\sigma = 0.5$ |
|---|---|---|---|---|
| Sequence + Structure | BA | 2.57e-06 | 3.48e-06 | 3.37e-06 |
| Sequence + Structure | EL | 1.15e-05 | 1.85e-05 | 1.97e-05 |
| Structure-only | BA | 8.54e-04 | 1.6e-03 | 2.3e-03 |
| Structure-only | EL | 0.016 | 0.051 | 0.069 |

## D.3 PERFORMANCE ACROSS MHC-II ALLELES

We further evaluate the performance across different MHC-II alleles. The results from our best peptide model is shown in Table A4. The best antigen-based model has an average CR-AUC score of 0.6649, with MHC-II specific scores of DP = 0.721, DQ = 0.598, and DR = 0.612. In general, DQ has the lowest performance, followed by DR and DP. This could be attributed to the uneven distribution of samples across MHC-II types. In training data, both EL and BA datasets have highly unbalanced MHC-II coverage, with DP:DQ:DR ratio equals 16:8:76 and 42:20:38, respectively. The BA test set is also unbalanced with only 5% of samples from DQ and DP, making their performance less reliable. This imbalance is inevitable, as the latest binding affinity entries in IEDB after 2020, as our initial test candidates, are already heavily skewed towards DR, which accounts for 95% of samples. One reason for this bias is that DR alleles are often expressed at higher levels on antigen-presenting cells, making them more dominant in immune presentation and easier to study experimentally. In contrast, the MHC-II distribution on EL test set is much more balanced (38% DR, 42% DP, and 20% DQ), offering a reliable view of how models perform across MHC-II types. In short, the smaller number of DQ allele samples may be the source of challenges behind achieving good model performance.

## D.4 SENSITIVITY ANALYSIS WITH STRUCTURAL NOISE

Since the predicted structure from AlphaFold3 may suffer from errors that propagate to the main model, we perform a sensitivity analysis of the model's outputs against structural noise. We first train a model variant that only takes MHC-II structures as inputs instead of both MHC-II sequences and structures. Note that the amino acid type information is inherently encoded in the structure. We then perform a sensitivity analysis by evaluating output variance under settings of simulated structure prediction errors. This is achieved via structure perturbation. Gaussian noises with mean 0 and three base scales, $\sigma \in \{0.1, 0.3, 0.5\}$, are added to the atom coordinates. To mimic the actual prediction error, the scale is further weighted by the pLDDT score (ranging from 0 to 100) of each atom, which is a confidence estimate from AF3. The less confident the prediction, the more noise is added to the structure. Concretely, noise is sampled from $\mathcal{N}(0, \sigma(1 - pLDDT/100))$. We generate 5 perturbed structures for each MHC-II and base scales, and convert them into the input structure tokens. We then report the output variance averaged across all peptide-MHC-II test pairs in BA and EL in Table A5. The outputs are highly stable for the sequence-structure model. This is expected since the sequence modality is more robust to noise or prediction errors. On the other hand, the structure-only model shows much larger output variance as the noise increases, despite its comparable performance. This indicates the advantages of explicitly integrating sequence information as a separate modality for robust prediction in realistic and noisy settings.

## D.5 QUALITATIVE ANALYSIS OF CR-AUC

To better understand what CR-AUC captures and the outcome difference between peptide-based and antigen-based models, we select three typical antigen-MHC-II pairs from the test set and visualize their coverage-redundancy (CR) curve as shown in Figure A3. All examples suggest that antigen-

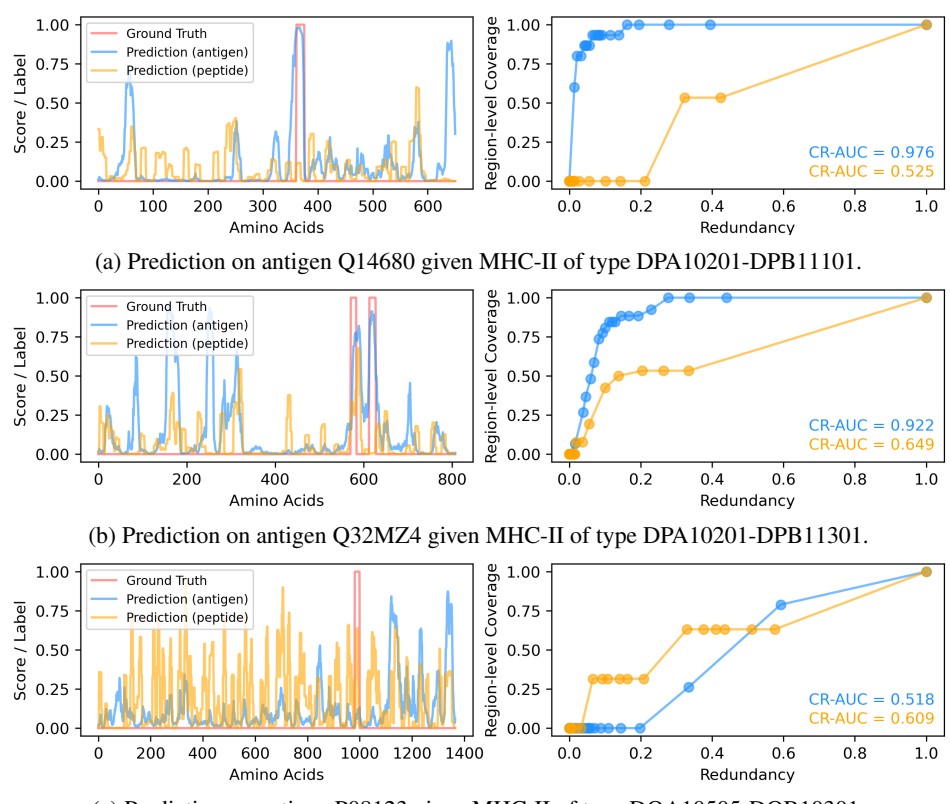

(a) Prediction on antigen Q14680 given MHC-II of type DPA10201-DPB11101.

(b) Prediction on antigen Q32MZ4 given MHC-II of type DPA10201-DPB11301.

(c) Prediction on antigen P08123 given MHC-II of type DQA10505-DQB10301.

Figure A3: Example performance comparison between peptide-based models and antigen-based models on two antigen proteins using coverage-redundancy curve. The blue and orange line in the left plots indicate the predicted residue-level scores, while the red line captures the ground truth regions.

based models are more likely to produce localized and confident predictions along the antigen sequence, given its richer context. Figure A3b represents cases where the same antigen contains multiple observed epitopes. Since antigen-based models can capture of them, it results in a steeper CR curve with higher CR-AUC value. Conversely, Figure A3c presents a case where the antigen-based model fails to detect the observed epitope. Even through the peptide-based model successfully identifies the epitope, it generates a lot more region proposals, resulting in a flatter CR curve compared to Figure A3a.

# E  COMPUTATIONAL RESOURCES

All experiments in this work were conducted on an A6000 GPU. Using the training hyperparameters described above, one round of joint BA and EL training takes approximately 20 hours to complete on an Intel(R) Xeon(R) w7-2495X CPU, while one round of antigen training takes approximately 4 hours to finish. The primary computational bottleneck is I/O speed, as each training iteration requires access to the huge precomputed ESM2 database (207GB in total).

# F  LICENSES FOR EXISTING ASSETS

Our dataset is mainly curated from IEDB (Vita et al., 2018), which is funded by National Institute of Allergy and Infectious Diseases (NIAID). According to IEDB's copy right information, NIAID does not impose any restrictions on the use or distribution of data within IEDB. The other sources of MixMHC2pred2 (Racle et al., 2023), NetMHCIIpan-3.2 (Jensen et al., 2018), and NetMHCIIpan-4.0 (Reynisson et al., 2020) are all under the CC BY-NC 4.0 license.

Table A6: Basic statistics of our curated datasets. # denotes the count of unique objects. *Seq* refers to peptide sequences for peptide-level tasks and antigen sequences for antigen-level task. *Seq* ☉ indicates sequences that are presented in training. The test pairs are guaranteed to be unseen. The exact count for 1.1M and 0.9M are 1,113,537 and 897,984, respectively.

| | Peptide Binding Affinity | | | Peptide Presentation | | | Antigen Presentation | | |
|---|---|---|---|---|---|---|---|---|---|
| | Train | Val | Test | Train | Val | Test | Train | Val | Test |
| *#Pair* | 133,044 | 7,040 | 938 | 1.1M | 54,351 | 2,929 | 46,539 | 3,058 | 1,759 |
| *#Seq* | 16,946 | 800 | 196 | 0.9M | 52,467 | 2,414 | 9,200 | 2,041 | 1,382 |
| *#Seq* ☉ | - | 200 | 0 | - | 12,387 | 0 | - | 1,590 | 979 |
| *#MHCII* | 77 | 60 | 28 | 132 | 83 | 72 | 121 | 57 | 73 |

Table A7: Mapping of evaluation methods (column) and benchmark tasks (row).

| Scale | Peptide-level Tasks | | Antigen-level Task |
|---|---|---|---|
| | Binding Affinity | Eluted Ligand | Eluted Ligand |
| Peptide-level | RMSE, AUC | Accuracy | - |
| Epitope-level | FRANK, $AUC_{epitope}$ | FRANK, $AUC_{epitope}$ | - |
| Antigen-level | - | CR-AUC | CR-AUC |

The motif deconvolution software MoDec (Racle et al., 2019b) employs a custom software license for academic non-commercial research purposes only. AlphaFold3 (Abramson et al., 2024) is licensed under CC BY-NC-SA 4.0.

# G    USE OF LARGE LANGUAGE MODELS (LLMS)

We only use the LLMs to correct the grammar and polish the writing in this work.

Table A8: The full results table of Table 3 in the main text. ↓ means lower is better, and vice versa.

| Input | | Strategy | | Binding Affinity | | | | Eluted Ligand | | | |
| ESM2 | Struct | Joint | Aux | RMSE↓ | AUC↑ | FRANK↓ | AUC$_{epitope}$↑ | Accuracy↑ | FRANK↓ | AUC$_{epitope}$↑ | CR-AUC↑ |
|---|---|---|---|---|---|---|---|---|---|---|---|
| ✓ | | | | 0.2553 | 0.7547 | 0.2240 | 0.7717 | 0.6470 | 0.1660 | 0.8253 | 0.6048 |
| ✓ | ✓ | | | 0.2466 | 0.7313 | 0.2425 | 0.7615 | 0.6098 | 0.1850 | 0.8095 | 0.6101 |
| ✓ | | ✓ | | 0.2584 | 0.7367 | 0.2401 | 0.7564 | 0.6582 | 0.1642 | 0.8264 | 0.6198 |
| ✓ | ✓ | ✓ | | 0.2553 | 0.7473 | 0.2374 | 0.7747 | 0.6354 | 0.1587 | 0.8328 | 0.6045 |
| ✓ | ✓ | ✓ | | 0.2419 | **0.7656** | 0.2134 | 0.7658 | 0.6763 | 0.1457 | 0.8372 | 0.6420 |
| ✓ | ✓ | ✓ | ✓ | **0.2408** | 0.7627 | **0.1855** | **0.8127** | **0.6955** | **0.1409** | **0.8492** | **0.6634** |

Table A9: The full results table of Table 4 in the main text. ↓ means lower is better, and vice versa.

| Method | Binding Affinity | | | | Eluted Ligand* | | | |
| --- | --- | --- | --- | --- | --- | --- | --- | --- |
| | RMSE ↓ | AUC ↑ | FRANK ↓ | $AUC_{epitope}$ ↑ | Accuracy ↑ | FRANK ↓ | $AUC_{epitope}$ ↑ | CR-AUC ↑ |
| NetMHCIIPan4.3 (Nilsson et al., 2023a) | **0.2295** | **0.8115** | 0.1886 | 0.8236 | 0.4980 | 0.1292 | **0.8672** | 0.6526 |
| RPEMHC (Wang et al., 2024) | 0.2433 | 0.7978 | **0.1569** | **0.8436** | - | - | - | - |
| MixMHC2Pred2 (Racle et al., 2023) | - | - | - | - | 0.3462 | 0.1420 | 0.8658 | 0.6906 |
| ImmuScope (Shen et al., 2025) | - | - | - | - | 0.6570 | 0.1240 | 0.8549 | 0.6796 |
| Ours | 0.2408 | 0.7627 | 0.1855 | 0.8127 | **0.7347** | **0.1192** | 0.8662 | **0.7349** |

