# OpenReview forum: "Accelerating MHC-II Epitope Discovery via Multi-Scale Prediction in Antigen Presentation"
_ICLR.cc/2026/Conference — Submitted to ICLR 2026_

### Official Review · Reviewer_ZWfD · 2025-10-17

**Soundness:** 3
**Presentation:** 2
**Contribution:** 4
**Rating:** 6
**Confidence:** 3

**Summary:**

This paper presents a large and standardized human MHC-II dataset encompassing peptide binding affinity, peptide presentation, and a novel antigen-level presentation task with antigen annotations. It introduces a multi-scale evaluation framework featuring a new coverage-redundancy AUC (CR-AUC) metric for antigen localization. The proposed sequence and structure aware models, utilizing ESM2 embeddings and predicted MHC-II structures, are evaluated through ablation studies. Results demonstrate improvements in antigen-level localization and competitive peptide-level performance against recent baselines.

**Strengths:**

The adaptive immune system is highly complex, and CD4+ T cells play a central role, yet predicting their behavior remains challenging. By framing this problem for the machine learning community, the work opens opportunities for novel computational approaches and insights. Additionally, computational studies in this area often rely on different datasets and metrics, making method-level comparisons difficult. This work addresses this issue by curating a standardized dataset and clearly defining tasks with corresponding evaluation metrics, enabling rigorous and fair comparisons across methods.

**Weaknesses:**

The main weaknesses of this paper is clarity. The definitions of evaluation metrics and associated notations are not clearly presented, and key methodological details are deferred to the appendix. While the authors propose a new method, as a dataset and benchmark paper, the focus should primarily be on the dataset and existing methods. The proposed method could constitute a separate manuscript. The usage of baselines within the benchmark is also not clearly described. From a technical perspective, important metrics such as generalization ability are missing from the evaluation. Additionally, the performance table shows minor differences between results, but no statistical significance tests are reported.

**Questions:**

1. **Figure 1**: There is a red bar in the third space. What does it represent?
2. **Section 3.4**: In the data enrichment section, the authors mention using ESM2 embeddings as pseudo-labels. If this refers to the proposed model in the appendix, this section should be rephrased to clearly explain how pseudo-labeling works.
3. **Line 334 and 355**: Please define $\mathbb{1}$ (the double-stroked "1").
4. **Line 350**: Please define $w_n$ and $w_{n'}$.
5. **Table 4**: The "ours" model is compared in Table 4 but described only in the appendix, which may confuse readers. As this is primarily a dataset paper, the new method could be better presented as a separate manuscript.
6. **Tables 3 and 4**: Some differences (e.g., AUC for BA and AUC epitope for Eluted Ligand) are minor. Including statistical significance tests would strengthen the analysis.
7. Generalization is an important metric for this task. Adding a holdout evaluation would be valuable.
8. Characterizing how antigens with annotations differ from the full antigenome would be valuable.
9. Were all baselines (NetMHCIIpan4.3, MixMHC2Pred2, RPEMHC, ImmuScope) re-run on this benchmark, or were pretrained weights used? Please clarify how peptide length and amino-acid constraints were handled for each baseline.
10. (*Optional*) For the antigen-level annotation and presentation task, is it possible to incorporate structure-aware antigen processing prediction methods (e.g., APL [Charles, 2022, Biochem.] and GPUCOREX [Li, 2024, BIBM]) using AlphaFold-predicted structures?
11. (*Optional*) Since many biological researchers still use legacy tools such as NNAlign provided by IEDB, including additional baselines covering these commonly used methods could make the benchmark more relevant and encourage the community to adopt newer approaches.

---

> ### Author Response · Authors · 2025-11-24
>
> **General response to the main story line and paper clarity:**
>
> Thank you for your valuable feedback. We now realize the clarity issues in our presentation and have revised the manuscript accordingly. As you noted, this paper is intended as a dataset and benchmark paper instead of proposing any new model for this problem. The dataset we curated contains two components of peptide-MHCII dataset and antigen-MHCII dataset. We expand and standardize the existing peptide-MHCII datasets. The antigen-MHCII dataset is curated for a novel task of antigen-MHCII prediction. This task subsequently requires a new method accordingly. However, we do not claim that antigen-based modeling is a better approach to study the MHCII presentation pathway. Our goal is simply to demonstrate that ML should address both peptide and antigen tasks, as they represent different biological scales of MHCII presentation (as shown in Figure A2).
>
> The reason why we ran a bunch of baseline methods is to demonstrate the state-of-the-art performance as a reference for future ML development. We also built our own architectural framework for the purpose of comprehensive benchmark analysis over different modeling designs (e.g., use of structural features, ESM2 embedding) instead of proposing a new model.
> That said, we purposely make our model architecture as modular and flexible as possible to accommodate various design choices. Since there exists no antigen-based model, there is no state-of-the-art method to compare with. However, antigen CR-AUC scores computed from existing peptide-based methods can still serve as a rough comparison of state-of-the-art performance in antigen-MHCII tasks. In the updated manuscript, we have made several revisions to clarify the purpose of our architectural framework and the role of the baseline experiments.
>
>
> **Response to “ours” model in Table 4:**
>
> The “ours” model in Table 4 corresponds to our architectural framework trained using the best configuration in Table 3, whereas Appendix Table A2 only serves as an additional ablation study to examine how different encoders affect performance. In the updated manuscript, we have added a brief summary of our “best” model architecture at the beginning of Section 5, with full implementation details provided in Appendix. For improved clarity, we also added Figure 2, which provides an overview of our experimental framework.
>
> Meanwhile, we would like to emphasize again that this model is not intended as a new method. Instead, it mainly represents the architectural framework we use for the benchmark analysis using the best configuration we found.

---

> ### Author Response · Authors · 2025-11-24
>
> **Response to metrics for model generalization ability:**
>
> The test set we constructed represents the holdout set to evaluate model generalizability. As described in section 3.2, we carefully curate the test set under several strict constraints. For the peptide-MHCII test set, not only are the peptide sequences entirely unseen from training, all the  9-mers (subsequences of length 9, which is the length of binding core) within testing peptides are also unseen from training. This constraint is standard in the immunotherapy literature (e.g., NetMHCIIpan) for assessing generalizability of peptide-MHCII models, yet the test sets curated by many existing works failed to satisfy it. For the antigen–MHCII test set, the epitopes (positive peptides) within each antigen are considered under the same 9-mer constraint to ensure that generalizability is properly evaluated.
>
>
> **Response to the red bar in Figure 1:**
>
> As written in section 3.3, the red bar highlights the raw label statistics of the DRB10101 MHCII type. We apologize for not including this annotation in the figure's caption, and we have adjusted accordingly in our updated manuscripts.
>
>
> **Response to the pseudo-labels in section 3.4:**
>
> We apologize for the confusion because of our unclear explanation. The data enrichment includes the enrichment of both input features and output labels, which is clarified in the updated manuscript. ESM2 embedding belongs to the enrichment of input features, as explained in the input configuration part of section 5.1. Only the estimated binding core is used as pseudo labels of an auxiliary task. The detailed implementation is included in the training strategy part of section 5.1, where we used a 2D convolution prediction head on top the cross attention weights to predict the estimated core.
>
>
> **Response to the ambiguous notation w_n, w_n’, and 1():**
>
> We apologize for the ambiguous notation used in our original explanation of CR-AUC. The double-stroked “1” represents the indicator function. The subscripts $n$, $n'$, and $m$ were intended to represent indices ($n$ and $n'$ indexing $G$, and $m$ indexing $P$). Our initial aim was to differentiate them from $i$ and $j$, which index residues and epitopes, respectively. In Equation (2), both $w_n$ and $w_{n'}$ refer to the unnormalized weights of $G$ for coverage computation, which differs only in their indices. However, we now realize that this notation may cause more confusion. To improve clarity, we have made the following adjustments in the updated manuscript:
>
> 1. All indexing subscripts $n$ and $m$ have been replaced with $i$ and $j$. The symbols $n$ and $m$ now refer only to the total number of ground truth regions $G$ and predicted regions $P$, respectively.
>
> 2. We also replaced $|G_n \cap P_m|$ in Equation (1) with $\sum_j |G_i \cap P_j|$, which more clearly represents the total number of positive predictions within $G_i$.
>
> We hope these updates make the notation clearer, and please let us know if there is any other ambiguity.

---

> ### Author Response · Authors · 2025-11-24
>
> **Response to detailed experimental setup of baseline models:**
>
> The performance of all baseline methods in Table 4 is obtained directly from their publicly released models. The NetMHCIIpan family and MixMHC2Pred2 only provide precompiled models and cannot be retrained. The RPEMHC and ImmuScope are retrainable, and we added a retrained version in our updated manuscript. In order to only provide insight about the key architectural difference between our architectural setup and theirs, we retrained our own replication of their models. In the updated Table 4, Ours_{RPEMHC} replaces the peptide-MHCII cross-attention module with a 2D convolution over pairwise residue features, while Ours_{ImmuScope} augments our model with additional convolutional refinement blocks for peptide representations after cross-attention. Other than the architectural difference, all other settings are the same as Ours. We have introduced more discussion regarding these new results in section 5.2 of the updated manuscript.
>
> Regarding constraints of peptide length and amino acid type, only MixMHC2Pred2 imposes strict limitations on its input. It only accepts peptides of length 12–21 and only supports peptides composed of the 20 canonical amino acids. For example, sequences containing selenocysteine (U) cannot be processed by MixMHC2Pred2. To ensure a fair comparison, we therefore filter the test set to contain only peptides that MixMHC2Pred2 can handle and report all baseline performance on this subset in Table 4.
>
> **Response to the possible use of structure-aware antigen processing prediction:**
>
> It would be infeasible to utilize peptide-MHCII complex structure from AlphaFold3 in our case. We compute and utilize the predicted holo (unbound) structures of MHCII in our experiment because the number of MHCII alleles is finite in the human body, these structures can be cached and reused repeatedly for any peptide-MHCII pairs. However, the total number of unique peptide-MHCII pairs goes up to millions. AlphaFold3 is computationally expensive with the computation of MSAs (multiple sequence alignment) as its main bottleneck. Based on our optimized multi-processing AF3 inference pipeline, the structure prediction takes about ~2 minutes for each peptide-MHCII pair. With over 1 million unique peptide-MHC-II pairs in our dataset, structure generation would take ~3.81 years on a single GPU. Moreover, inference time complexity also becomes significant since complex structures need to be prepared across all candidate peptides within the antigen across available MHC-II.
>
>
> **Response to the consideration of NNAlign:**
>
> Thank you for pointing this out. In fact, NNAlign is the exact backbone architecture underlying the NetMHCIIPan family. In other words, our benchmark results already cover the most widely used and state-of-the-art approaches in the community.

---

> ### Comment · Reviewer_ZWfD · 2025-11-24
>
> Thank you for the clarification. Your revisions addressed my concerns on presentation and clarity. I have updated the presentation score to 3 and raised my overall rating to 8.
>
> I have two remaining points that would benefit from further clarification:
>
> 1. Modular architecture:
> Since the manuscript emphasizes the dataset and modular design, it would be helpful if you could share a small code/documentation fragment (or a brief supplementary file) to illustrate how the modular components are implemented. This would strengthen confidence that the framework will support future model development.
> 2. Baseline coverage:
> While you compare against state-of-the-art models, biological researchers often rely on multiple NNAlign and NetMHCIIpan versions due to differences in data and training strategy. Although I do not expect this work to include all such comparisons, it would be useful to discuss this point in the future repository README.
>
> The two points above are my only remaining concerns.
>
> Overall, it is a good paper from my perspective, and no apology is needed. Thank you for addressing my concerns. I hope your research continues to progress well.

---

### Official Review · Reviewer_YXiH · 2025-10-31

**Soundness:** 2
**Presentation:** 2
**Contribution:** 2
**Rating:** 2
**Confidence:** 4

**Summary:**

This paper aims to model the MHC-II antigen presentation pathway from processing, binding, to presentation, which requires the effective integration of biological context information, binding affinity measurements, and eluted ligand data. To capture peptide–MHC interactions, the authors employed a cross-attention mechanism between the MHC molecule and the peptide, and identified several key factors that contribute to improved predictive performance, including the use of ESM-derived representations, MHC structural features, joint learning, and auxiliary binding-core prediction. To further enrich contextual modeling, the authors extended the prediction scope from peptide level to antigen-protein level, leveraging source protein data collected from IEDB for model training. This extension demonstrates significantly enhanced prediction accuracy compared with conventional peptide-based approaches.

**Strengths:**

1. Modeling MHC-II antigen presentation directly from the antigen-protein perspective is an interesting and meaningful research direction.

2. The combination of MHC ESM embeddings and structural features effectively enhances predictive performance.

3. The description of the dataset and modeling details is clear, well-structured, and easy to follow.

**Weaknesses:**

1. The novelty of the work is limited.
a) The proposed network architecture is highly similar to ImmuScope, with only minor modifications.
b) The use of joint training (as in NetMHCpan-4.0), binding core prediction (NetMHCIIpan-3.2), ESM embeddings (HLAIIpolo, Pep2Vec), and structural features (NetMHCpan-4.2) for performance enhancement is already well-established in the field of MHC-II peptide presentation.
c) Moreover, recent versions of NetMHCIIpan (≥4.0) have already incorporated peptide context information, which has been shown to substantially improve presentation prediction performance, further diminishing the claimed novelty of this study.

2. The experimental design is insufficient and potentially biased.
a) Since antigen presentation is the central focus of this paper, Table 5 lacks comparative experiments with existing methods; at minimum, results from peptide-based baselines should be included for fair comparison.
b) Additionally, the peptide-based prediction appears to be conducted using the entire protein sequence, whereas the antigen-based prediction is restricted to k-length fragments of the protein, which artificially reduces the search space and may lead to unfair performance advantages.
c) In addition to the lack of T-cell data and multi-allele samples, the authors could include more experiments to demonstrate that their dataset offers better quality compared with existing BA and EL datasets, rather than only being a combination of multiple sources.

**Questions:**

Major
1. The authors should include a diagram of the model architecture to clearly illustrate the framework to readers, including the key components for both peptide-level and antigen-level presentation modeling. Moreover, based on the textual description, the architecture appears very similar to ImmuScope encoder. The authors must explicitly highlight the creative differences and methodological innovations that distinguish their work from ImmuScope.

2. In Tables 3–5, the accuracy metric appears to be biased, as the threshold is fixed at 0.5. In contrast, NetMHCIIpan reports a top-ranked percentile score, which is normalized and therefore may provide a more appropriate decision threshold (top-20% or top-30%) than a fixed 0.5 cutoff.

3. In Table 5, additional comparisons with other peptide-based methods are necessary, especially considering the NetMHCIIpan-4.3 with context information used.

4. The peptide-based prediction appears to use the full-length protein sequence, applying a fixed 9-mer sliding window. In contrast, the antigen-based prediction seems to be conducted on k-length fragments, which reduces the effective search space and may artificially inflate the reported accuracy.

5. In the Method Comparison section, it is unclear whether the other models were retrained on the same dataset or evaluated using their original parameters. If they were not retrained, could this affect the fairness of the comparison? Additional clarification is needed.

Minor
1. In Table 1, the data statistics are inconsistent with the textual description. For instance, the text mentions 141K BA pairs covering 78 MHC-II molecules, whereas Table 1 reports 136K pairs and 77 MHC-IIs. Please verify and ensure consistency between the text and tables.

2. The use of highly complex features, such as ESM and structure features, may negatively affect computational efficiency. It is recommended to include a running time to demonstrate the model efficiency.

---

> ### Author Response · Authors · 2025-11-24
>
> **Clarification:**
>
> Before addressing the specific concerns in the reviews, we would like to first clarify several misunderstandings about NetMHCIIpan and why it cannot provide the modeling insight shown by our work.
>
> 1. The NetMHCIIpan family did not incorporate binding core prediction in training. Although the papers discuss the performance of predicted binding cores, their models simply treat the core as a short peptide sequence and predict its EL presentation at the inference stage. This is fundamentally different from our auxiliary supervision of identifying the binding core given peptide-MHCII pair.
>
> 2. We also did not find evidence supporting the assertion that “structural features (NetMHCIIpan-4.2) for performance enhancement are already well-established.” The NetMHCIIpan models are sequence-based and do not incorporate any explicit 3D structural features in their architecture. Their papers did not discuss any ML experiments relevant to structural features.
>
> 3. While NetMHCIIpan (≥4.0) has the option of context encoding, their context is defined narrowly as three upstream and three downstream flanking residues (six neighboring residues in total). This provides limited sequence context and remains fundamentally different from our antigen-MHCII formulation, in which the model receives the full antigen sequence and learns to identify biologically relevant regions involved in MHCII presentation. However, for completeness, we have updated our benchmark results to include the context-encoding variant of NetMHCIIpan 4.3.

---

> ### Author Response · Authors · 2025-11-24
>
> **Response to the novelty concern and main contribution:**
>
> We apologize for any confusion caused by the initial manuscript, and we would like to clarify the main goal and contribution of our work.
>
> This is a dataset and benchmark paper, with the main contribution being the curated dataset, diverse modeling tasks, designated test sets, comprehensive evaluation framework, and in-depth benchmark analysis for MHC-II presentation prediction. That said, we do not aim to propose any novel model architecture. Instead, we purposely make our architectural framework to be modular and adaptable for different ML modeling design choices (e.g., different architectures, input features, training strategies). Therefore, our benchmark experiments mainly serve to (1) establish strong baseline performance for the proposed tasks and (2) provide modeling insights in how different ML design choices affect the performance. To the best of our knowledge, no prior work has performed such a systematic ML analysis like us.
>
> We are not presenting the antigen presentation as the central focus of this paper. Our dataset contains two components of peptide-MHCII dataset and antigen-MHCII dataset. While the ML task of peptide-MHCII is already established, we propose a new antigen-MHCII task based on the curated antigen-MHCII samples. It subsequently requires the models to be capable of identifying potential epitopes from antigen sequences. As shown in Figure A2, peptide-MHCII EL and antigen-MHCII EL tasks capture the MHCII presentation pathway at different biological scales. We are claiming that the researcher should develop models for both peptide tasks and antigen tasks. We are not promoting antigen modeling as a better approach to study the MHCII presentation pathway.
>
> We now realize how the paper presentation may mislead the audience to feel like we are proposing a new model for the problem. In the updated manuscript, we have made several revisions (Abstract, Introduction and Experiment) to clarify the role of model architecture and benchmark experiments in this work. We have also included Figure 2 as an overview of the architectural framework used in our experiments. As mentioned above, our primary goal is to offer a strong baseline performance and a rigorous benchmark analysis that future ML development can build upon.
>
>
> **Response to more comparative results of antigen modeling:**
>
> As mentioned above, the context encoding defined in NetMHCPan4.3 is fundamentally different from our antigen-MHCII formulation, while we have included its performance in the updated Table 4 for reference.
>
> Although there are no prior antigen-based models for direct baseline comparison, we conduct antigen-level evaluations on peptide-based models to enable an approximate comparison. In the updated Table 5, we include a new test result (named as “random*”) from our antigen-based model, where the test data is filtered according to MixMHC2Pred2’s constraints (as in Table 4). This makes the CR-AUC score strictly comparable to all other CR-AUC scores in Table 4. The original (“random”) CR-AUC score is strictly comparable to those in Table 3.
>
> Our best antigen-based model outperforms most of the prior peptide-based models, but falls slightly behind our best peptide-based model with joint training and auxiliary core prediction. Surprisingly, we find that NetMHCIIpan4.3 with context encoding performs worse than expected. However, since the original paper provides limited information on how the context encoding is implemented, trained or evaluated, we are unable to determine the cause of this drop in performance.
>
> **Response to baseline computation details and comparative results:**
>
> The baseline performance of prior models in Table 4 is measured using their publically released models. Both NetMHCIIPan4.3 and MixMHC2Pred2 have their models precompiled and cannot be retrained. The RPEMHC and ImmuScope are retrainable, and we have added a retrained version in our updated manuscript. In order to only provide insights about the key architectural difference between our architectural setup and theirs, we retrained our own replication of their models. In the updated Table 4, Ours_{RPEMHC} replaces the peptide-MHCII cross-attention module with a 2D convolution over pairwise residue features, while Ours_{ImmuScope} augments our model with additional convolutional refinement blocks for peptide representations after cross-attention. The rest of the settings (e.g., input features, training strategies) remain the same for the two variants for fair comparison.
>
> Our best model performs slightly better than Ours_{RPEMHC} on average, which is expected since 2D convolution is less efficient than cross-attention at capturing global interactions. Ours_{ImmuScope} shows improvement on BA specifically. Since BA signals are more sensitive to the binding core, the additional convolutional refinement may help the model focus on the most relevant local regions for prediction.

---

> ### Author Response · Authors · 2025-11-24
>
> **Response to the “biased” accuracy metric for NetMHCIIpan:**
>
> We apologize for not including enough details about the computation of the baseline performance. The accuracy metric is not biased for NetMHCIIpan. NetMHCIIpan is able to output both the raw predicted scores of peptide BA and EL, as well as a separated percentile rank score. In fact, MixMHC2Pred2 can also output both the raw scores and the percentile score. The performance in Table 4 is evaluated using only their raw scores for fair comparison. The reason why we exclude the percentile score is because of its lack of implementation consistency. Percentile rank scores is computed by comparing the current predicted score to a distribution of predicted scores of a predefined set of random natural peptides. We can only disclose from NetMHCIIpan3.2’s web server that their set contains 200,000 random natural peptides, while MixMHC2Pred2 utilized a set of 10^6 random human peptides to compute its percentile score. Given that our models are trained with the same supervision signal of BA (normalized IC50) and EL (binary classification), we believe the raw scores are comparable and the accuracy metric is unbiased.
>
>
> **Response to the concern of different sequence length for peptide and antigen prediction:**
>
> We apologize for the confusion in our initial manuscript. Because peptide-based models can only provide per-peptide scores, we need to apply a sliding window (size 9 as the core size) over the antigen to score all possible peptides and then aggregate these scores into per-residue scores for CR-AUC evaluation. For antigen-based models, the k-length truncation only happens at training. The original motivation was to prevent out-of-memory errors when handling very long antigen sequences (e.g., length >1000) in batched training. This is the same as AlphaFold2, which crops proteins to lengths of 256/384 during training. We later found that varying k also provides additional supervision signals for antigen-based models. However, the evaluation is performed using the full antigen sequence as input, which means that the reported performance is not biased or inflated.
>
>
> **Response to “inconsistent” data statistics:**
>
> Table 1 only represents the data statistics of our training set, which is mainly used for statistical comparison of training sets used in other works. The full data statistics table, including both training, validation, and testing set, is shown in Table A6. The total number of BA pairs is indeed 141K. Due to the page limits, we put Table A6 in the Appendix.
>
>
> **Response to computational efficiency:**
>
> Since most peptides in the peptide-MHCII setting are only ~15 residues long, computing ESM2 embeddings is lightweight. In our experiment, generating embeddings for one million peptides can be done within an hour. The main computational cost lies in obtaining the 3D structures of MHCII via AlphaFold3. Fortunately, the number of human MHC-II alleles is finite, so their structures only need to be predicted once and can then be cached for repeated use. Additionally, we apply the structure quantization method (ProSST) to convert full-atom 3D structures into structural token sequences (matching the length of amino-acid sequence), which further reduces downstream modeling complexity. We will release both the precomputed MHC-II structures and their corresponding structural tokens alongside the dataset upon acceptance.

---

### Official Review · Reviewer_QiFU · 2025-11-01

**Soundness:** 4
**Presentation:** 4
**Contribution:** 3
**Rating:** 6
**Confidence:** 4

**Summary:**

The authors collect a large dataset of 1.2M peptides interacting with 134 unique human MHC-II. This is certainly useful for the community, since many researchers are now attempting to predict immunological properties (such as antigenic immunogenicity) with data-hungry machine learning methods.

Although certainly very valuable, my only concern is whether ICLR is the best venue for this paper. Modeling contributions are minor: the architecture is standard (sequence encoders + cross-attention) and benchmarked against known baselines. Their model performs comparably, or marginally better.

**Strengths:**

The authors collect a large dataset of 1.2M peptides with 134 unique human MHC-II, labeled as positives or negatives depending on the interaction and whether the peptide is presented or not. This is certainly a welcome and valuable resource. They introduce an antigen-level prediction task and evaluation framework.

They enrich the dataset with annotations from multiple sources, such as ESM2 residue embeddings, predicted peptide binding motifs from MoDec, inferred structures by AlphaFold3, and others. These annotations will surely facilitate downstream work by other researchers.

They present a model of standard architecture (sequence encoders + cross-attention) trained on their dataset, and evaluated on 3 prediction tasks. Their model performs comparably, or marginally better than other state-of-the-art approaches.

**Weaknesses:**

One issue that the authors discuss is that the data for some MHC is very unbalanced. They attempt to rebalance by generating negatives by ad hoc data augmentation techniques.

The paper has little representation learning content, so my only concern is whether ICLR is the best venue for this work. In fact, the model they use for predictions is only described in the Appendix, with little mention given in the main text.

**Questions:**

- Can the authors describe in more detail their model in the main text?
- Although the dataset is certainly valauable, I believe the authors should justify the novelty of their modeling approach in order to fit a venue such as ICLR.

---

> ### Author Response · Authors · 2025-11-24
>
> **Response to the concern of modeling novelty:**
>
> Thank you for recognizing the value of our curated dataset. As a dataset and benchmark paper, our primary goal is to bring the MHCII presentation problem to the broader ML community and bridge the gap between ML developed in this domain and the ML advances in general. This includes the proposal of a new dataset, a comprehensive evaluation framework, and a detailed benchmark study. That said, this work does not aim to propose a new model to study the MHCII presentation pathway, but rather to establish a rigorous benchmark analysis that future ML development can build upon.
>
> Our dataset consists of both peptide-MHCII samples and antigen-MHCII samples. While the ML task of peptide-MHCII is already established, we propose a new antigen-MHCII task based on the curated antigen-MHCII samples. It subsequently requires the models to be capable of identifying potential epitopes from antigen sequences. We are claiming that the researcher should develop models for both peptide tasks and antigen tasks.
>
> To serve as a benchmark study, we deliberately make our architectural framework to be modular and compatible with common ML components used in AI for science (e.g., structural features, ESM2 embeddings). This allows us to systematically analyze how different components influence performance (Table 3 and Table A2) and to offer practical insights for model development. We also evaluate several widely used models on our dataset to establish state-of-the-art performance as references (Table 4). On the other hand, since there exists no prior antigen-based models, there is no baseline to compare with. However, our proposed CR-AUC score enables a rough but informative comparison between peptide-based and antigen-based baselines to show that the antigen results are reasonable.
>
> In summary, since the main focus of this work is the dataset, task formulation, evaluation framework, and benchmark analysis, we believe the concerns about architectural novelty do not apply to the intended scope of this paper.
>
> **Response to more model details in the main text:**
>
> In the updated manuscript, we added Figure 2 that presents the architectural overview and a model summary at the beginning of section 5. However, due to page limits, the model details remain in the Appendix.

---

### Official Review · Reviewer_NMz9 · 2025-11-03

**Soundness:** 3
**Presentation:** 3
**Contribution:** 3
**Rating:** 6
**Confidence:** 4

**Summary:**

This paper presents a well-curated dataset for predicting MHC-II antigen presentation. The authors integrate and standardize data from IEDB and other public sources, introducing not only peptide-level but also antigen-level tasks for the first time. They propose a multi-scale evaluation framework and conduct experiments comparing various input features, model architectures, and training strategies. The antigen-based modeling approach shows promising results in localizing epitope regions within full antigen sequences, outperforming peptide-based models in region-level coverage-redundancy analysis.

**Strengths:**

This manuscript introduces antigen-level prediction for MHC-II, with a novel evaluation metric (CR-AUC). In addition, it has rigorous dataset curation, extensive experiments, and thorough ablation studies. The manuscript is well organized and clearly written. Furthermore, it provides a good resource for the community and advances the modeling of MHC-II antigen presentation.

**Weaknesses:**

The model underperforms in peptide-binding affinity (BA) prediction compared to state-of-the-art methods like RPEMHC and NetMHCIIpan4.3, an issue the authors attribute to checkpoint selection bias but which could be addressed more systematically. Furthermore, the study is limited to single-allele data and does not handle the more complex but common real-world scenario of multi-allelic mass spectrometry samples. The antigen-level modeling is also potentially impacted by missing antigen annotations for a subset of peptides, which may introduce bias. Most critically, the training labels (BA and eluted ligands) are only proxy signals for immunogenicity, and the models do not incorporate direct T-cell response data due to its scarcity, leaving a gap between prediction and actual immune recognition.

**Questions:**

1. Could authors explore alternative checkpoint selection strategies (e.g., task-specific early stopping) to improve BA performance without compromising EL results?
2. How might the missing antigen annotations affect model generalizability? Is there a way to quantify or mitigate this bias?
3. The structural inputs help in EL but not consistently in BA. Why is that? Could this be due to the smaller BA dataset? Would scaling the BA data help?

---

> ### Author Response · Authors · 2025-11-24
>
> **Response to the checkpoint selection strategies:**
>
> Thank you for raising this point. In our current implementation, we select checkpoints using the average of BA and EL validation AUCs for simplicity. However, in practice, we don’t necessarily need to use the same checkpoint for BA and EL prediction. In our experiments, we evaluate both tasks using a single shared checkpoint to remain consistent with the NetMHCIIpan family (they do not disclose how they balance performance between the two tasks).
> Upon quick experiment, we realized that if we use BA validation AUC alone for checkpoint selection, the model performance reaches a BA test AUC of 0.792 and an AUC_{epitope} of 0.8457, which is competitive with the other baseline methods. Its corresponding EL performance, on the other hand, is slightly compromised compared to best results in Table 3 (ACC = 0.6709; AUC_\epitope = 0.8325; CR-AUC = 0.6385). However, as mentioned above, users can simply choose the checkpoint optimized for their task of interest (BA or EL) in practice, ensuring that each task achieves its best possible performance.
>
> **Response to potential bias of missing antigen annotations:**
>
> We believe the concern about bias from missing antigen annotations does not directly apply to our setup. In this work, peptide-MHCII and antigen-MHCII are treated as two separate datasets corresponding to different stages of the MHCII presentation pathway (Figure A2). The two tasks are related but different, and we train and evaluate the models for the two tasks separately. When training on the peptide dataset, the model has no access to any antigen information regardless of whether it is available. When training on the antigen dataset, all samples have available antigen annotation. As a result, we believe the absence or presence of antigen annotation do not affect the model’s generalizability.
>
> Also, the availability of antigen information is mostly determined by whether the authors have provided the antigen information in the literature, or whether the original experiment begins with peptides versus full-length antigens. These factors are unrelated to binding patterns of peptide-MHCII. We find no evidence in the literature that antigen annotations are systematically more likely to be available for “easy” or “hard” binders.
>
>
> **Response to potential reasons for marginal improvement of structural inputs on BA:**
>
> We believe the smaller scale of the BA dataset is a reasonable explanation. It contains less samples compared to EL, which limits the model’s ability to fully capture the underlying structural patterns. This is also indirectly reflected by the results in Table A5, where the BA prediction is less sensitive to structural noise. It suggests that the model currently relies less on structural information when predicting BA compared to EL.
> Unfortunately, BA datasets are difficult to scale. We have already collected most of the available peptide-MHCII BA samples, and there is a lack of reliable computational strategies for augmenting BA data. Because BA is a continuous biochemical measurement of the binding strength between peptides and MHCII, even single-residue mutations or minor sequence extensions may cause unpredictable change.

---

> ### Author Response · Authors · 2025-11-24
>
> **Response to the absence of T-cell response data:**
>
> Several existing studies have demonstrated the strong correlation between EL results from mass spectrometry (MS) experiments and the downstream T-cell immune responses. For example, Alba et al. [2] have shown that more than 70% of MHC-II epitopes identified via MS (i.e., positive EL) from SARS-CoV-2-infected cells overlapped with experimentally verified CD4+ epitopes (i.e., positive immune responses). The same pattern was also discovered for influenza and allergy-related antigens [3]. Moreover, it is common in this field to directly use immune response data to validate the performance of EL prediction models [1, 4, 5], which indicates their connection.
>
> **Response to the concern of multi-allele EL data:**
>
> The current work only focuses on single-allele data. We intentionally exclude multi-allele data because it represents a different problem formulation where the supervision signal is fundamentally more ambiguous. Addressing the multi-allele setting requires a different data-processing pipeline, a different strategy for constructing data splits, and a tailored evaluation framework. To the best of our knowledge, existing studies only incorporate multi-allele data as an auxiliary signal to improve the performance of single-allele prediction. None of these works provide a systematic benchmark evaluation or architectural design that specifically targets the multi-allele task (a common practice is simply to use an attentive multi-instance learning module over multi-allele bags). Therefore, we believe the multi-allele problem should be comprehensively studied as a separate work, which is beyond the intended scope of this paper.
>
> **Reference:**
>
> [1] Shen et al., Self-iterative multiple-instance learning enables the prediction of CD4+ T cell immunogenic epitopes. Nat Mach Intell, 2025
>
> [2] Weingarten-Gabbay et al., The HLA-II immunopeptidome of SARS-CoV-2. Cell Rep, 2024
>
> [3] Wu et al., Quantification of epitope abundance reveals the effect of direct and cross-presentation on influenza CTL responses. Nat Commun, 2019
>
> [4] Racle et al., Machine learning predictions of MHC-II specificities reveal alternative binding mode of class II epitopes. Immunity, 2023
>
> [5] Nilsson et al., Accurate prediction of HLA class II antigen presentation across all loci using tailored data acquisition and refined machine learning. Sci Adv, 2023

---

### Author Response · Authors · 2025-12-01
**Rebuttal Summary for AC**

**Rebuttal Summary for AC**

We thank all reviewers for their diligent reviewing and valuable feedback, and we provide the following brief rebuttal summary for the AC’s reference. We sincerely appreciate the additional time and effort the AC has invested in assessing the review.

The main concern raised is the perceived lack of modeling novelty. We believe this stems from our unclear presentation of our work as a dataset-and-benchmark paper. The revised manuscript now clarifies our intended contributions and overall novelty.

**Reviewer QiFU** has the lowest initial rating of 2, whose concerns are mainly about (1) the novelty of the model architecture and (2) misunderstandings regarding existing literature and our experimental design.

In our rebuttal, we emphasize our contribution as a dataset and benchmark paper, with the goal of bridging the gap between methods developed for MHC-II presentation prediction and broader ML advances. Specifically, we propose multiple modeling tasks and a comprehensive evaluation framework based on our curated dataset, and perform an in-depth benchmark analysis using a modular experimental architecture. Our goal is not to introduce a new model architecture, but to provide strong baselines and actionable modeling insights for future ML development. To the best of our knowledge, no prior work has performed such a systematic ML-focused analysis in this problem. To avoid confusion for future readers, we have revised our manuscript to better explain our main contributions and the role of our own architectural framework within the benchmark analysis.

Regarding the concerns stemming from misunderstandings, we first clarify the misconception that existing methods like NetMHCIIpan already provide the same modeling insights. Several statements in the review are factually incorrect. We then address each specific issue related to model description, evaluation metrics and implementation, and detailed data statistics. Following the reviewer’s suggestions, we add a diagram of our architectural framework and include three additional baseline results in Table 4, along with more discussion in benchmark results. We believe all the concerns are fully addressed.

**Reviewer NMz9** (rating 6) raises additional discussion questions regarding checkpoint selection strategies, potential bias from missing antigen annotations, and performance differences when incorporating structural inputs. We believe we have provided sufficient discussion for each question. We also expanded our discussion of limitations related to T-cell immune response data and multi-allele data, which aligns with concerns raised in our initial limitation section.

**Reviewer QiFU** (rating 6) acknowledges the value of our curated dataset, but also raises concerns about modeling novelty. Same as above, our rebuttal first emphasizes contribution as a dataset and benchmark paper. We further explain that our architectural framework is purposely designed to be modular, as we aim to provide a benchmark analysis of common ML design choices used in AI-for-science to the proposed tasks. Such systematic analysis for this specific problem has been missing in prior work, and our work aims to fill that gap.

**Reviewer ZWfD** (rating 6) raises several clarity concerns regarding the role of our benchmark experiments and certain mathematical notations. It made us further realize how our initial manuscript may mislead the audience to think we are proposing a new method. In our rebuttal, we carefully address each issue arising from ambiguity in the initial paper presentation and revise the manuscript accordingly. We also clarify the misunderstanding about the held-out set, along with more implementation details about the baseline experiments. We are pleased that the reviewer found our responses and revisions satisfactory.

---

### Meta-Review · Area_Chair_KQKU · 2026-01-10

**Summary:**

This paper aggregates diverse sources into a standardized format is a helpful effort to the computational immunology community. The move toward antigen-level modeling addresses a critical stage of the presentation pathway often ignored by existing peptide-centric models. However, this paper has many weaknesses to be addressed:

1.	Lack of evidence for dataset quality improvement (Reviewer YXiH). As pointed out during the review process, the authors characterize this as a well-curated dataset. To justify a new dataset, authors should show that training on their curated version leads to better generalization or handles noise better than training on a simple aggregation of IEDB.

2.	This work introduces CR-AUC as a new metric to measure the trade-off between localization (coverage) and sparsity (redundancy). While the authors clarified the implementation details (inference vs. training truncation), they failed to address the more fundamental question of metric validity. There is no evidence in the paper or the rebuttal showing that CR-AUC correlates with practical utility in epitope identification.

3.	Absence of multi-allele EL and TCR data (Reviewer NMz9 and YXiH). The reliance on single-allele datasets oversimplifies the multi-allelic nature of the human immunopeptidome in vivo. AC agrees with the reviewers that a benchmark claiming to accelerate epitope discovery is incomplete if it sidesteps the primary challenge (multi-allele deconvolution) faced by the community.

4.	The authors' claim that existing methods lack similar modeling insights is contradicted by recent advancements in the NetMHC family. Specifically, Nilsson et al. (2025) introduced NetMHCpan-4.2, which explicitly integrates structural features and structural constraints of the MHC binding cleft to improve CD8+ epitope prediction. While the authors' work focuses on MHC-II, the modeling insight of leveraging structural information is not a unique conceptual proposition exclusive to this paper.

**Reviewer Concerns:**

Most concerns cannot be addressed.

**Reviewer Scores:**

Unlike to change.

---

### Decision · Program_Chairs · 2026-01-26

Reject